# Keep the Gradients Flowing: Using Gradient Flow to Study Sparse Network Optimization

## Abstract

Training sparse networks to converge to the same performance as dense neural architectures has proven to be elusive. Recent work suggests that initialization is the key. However, while this direction of research has had some success, focusing on initialization alone appears to be inadequate. In this paper, we take a broader view of training sparse networks and consider various choices made during training that might disadvantage sparse networks. We measure the gradient flow across different networks and datasets, and show that the default choices of optimizers, activation functions and regularizers used for dense networks can disadvantage sparse networks. Based upon these findings, we show that gradient flow in sparse networks can be improved by reconsidering aspects of the architecture design and the training regime. Our work suggests that initialization is only one piece of the puzzle and a wider view of tailoring optimization to sparse networks yields promising results.

## 1 Introduction

Over the last decade, a "bigger is better" race in the number of model parameters has gripped the field of machine learning (Amodei et al., 2018; Thompson et al., 2020), primarily driven by over-parameterized deep neural networks (DNNs). Additional parameters improve top-line metrics, but drive up the cost of training (Horowitz, 2014; Strubell et al., 2019; Hooker, 2020) and increase the latency and memory footprint at inference time (Warden & Situnayake, 2019; Samala et al., 2018; Lane & Warden, 2018). Moreover, overparameterized networks have been shown to be more prone to memorization (Zhang et al., 2016).

To address some of these limitations, there has been a renewed focus on compression techniques that preserve top-line performance while improving efficiency. A large amount of research focus has centered on pruning, where weights estimated to be unnecessary are removed from the network at the end of training (Louizos et al., 2017; Wen et al., 2016; Cun et al., 1990; Hassibi et al., 1993a; Ström, 1997; Hassibi et al., 1993b; Zhu & Gupta, 2017; See et al., 2016; Narang et al., 2017). Pruning has shown a remarkable ability to preserve top-line metrics of performance, even when removing the majority of weights (Hooker et al., 2019; Gale et al., 2019). However, most pruning techniques still require training a large, overparameterized model *before* pruning a subset of weights.

Due to the drawbacks of starting dense prior to introducing sparsity, there has been a recent focus on methods that allow networks which *start* sparse at initialization, to converge to similar performance as dense networks (Frankle & Carbin, 2018; Frankle et al., 2019b; Liu et al., 2018a). These efforts have focused disproportionately on trying to understand the properties of initial sparse weight distributions that allow for convergence. However, while this work has had some success, focusing on initialization alone has proven to be inadequate (Frankle et al., 2020; Evci et al., 2019).

In this work, we take a broader view of why training sparse networks to converge to the same performance as dense networks has proven to be elusive. We reconsider many of the basic building blocks of the training process and ask whether they disadvantage sparse networks or not. Our work focuses on the behaviour of networks with random, fixed sparsity at initialization and we aim to gain further intuition into how these networks learn. Furthermore, we provide tooling tailored to the analysis of these networks.

In order to effectively study sparse network optimization in a controlled environment, we propose an experimental framework, *Same Capacity Sparse vs Dense Comparison* (`SC-SDC`). Contrary to most prior work comparing sparse to dense networks, where overparameterized dense networks are compared to smaller sparse networks, `SC-SDC` compares sparse networks to their equivalent capacity dense networks (same number of active connections and depth). This ensures that the results are a direct result of sparse connections themselves and not due to having more or fewer weights (as is the case when comparing large, dense networks to smaller, sparse networks).

We go beyond simply comparing top-line metrics by also measuring the impact on gradient flow of each intervention. Historically, exploding and vanishing gradients were a common problem in neural networks (Hochreiter et al., 2001; Hochreiter, 1991; Bengio et al., 1994; Glorot & Bengio, 2010; Goodfellow et al., 2016). Recent work has suggested that poor gradient flow is an exacerbated issue in sparse networks (Wang et al., 2020; Evci et al., 2020). To accurately measure gradient flow in sparse networks, we propose a normalized measure of gradient flow, which we term *Effective Gradient Flow* (`EGF`) – this measure normalizes by the number of active weights and thus is better suited to studying the training dynamics of sparse networks. We use this measure in conjunction with `SC-SDC`, to see where sparse optimization fails and to consider where this failure could be a result of poor gradient flow.

**Contributions** Our contributions can be enumerated as follows:

1. **Measuring effective gradient flow** We conduct large scale experiments to evaluate the role of regularization, optimization and architecture choices on sparse models. We evaluate multiple datasets and architectures and propose a new measure of gradient flow, *Effective Gradient Flow* (`EGF`), that we show to be a stronger predictor of top-line metrics such as accuracy and loss than current gradient flow formulations.

2. **Batch normalization plays a disproportionate role in stabilizing sparse networks** We show that batch normalization is more important for sparse networks than it is for dense networks, which suggests that gradient instability is a key obstacle to starting sparse.

3. **Not all optimizers and regulizers are created equal** Weight decay and data augmentation can hurt sparse network optimization, particularly when used in conjunction with accelerating, adaptive optimization methods that use an exponentially decaying average of past squared gradients, such as Adam (Kingma & Ba, 2014) and RMSProp (Hinton et al., 2012). We show this is highly correlated to a high `EGF` (gradient flow) and how batch normalization helps stabilize `EGF`.

4. **Changing activation functions can benefit sparse networks** We benchmark a wide set of activation functions, specifically ReLU (Nair & Hinton, 2010) and non-saturating activation functions such as PReLU (He et al., 2015), ELU (Clevert et al., 2015), SReLU (Jin et al., 2015),Swish (Ramachandran et al., 2017) and Sigmoid (Neal, 1992). Our results show that when using adaptive optimization methods, Swish is a promising activation function, while when using stochastic gradient descent, PReLU preforms better than the other activation functions.

**Implications** Our work is timely as sparse training dynamics are poorly understood. Most training algorithms and methods have been developed to suit training dense networks. Our work provides insight into the nature of sparse optimization and suggests a wider viewpoint beyond initialization is necessary to converge sparse networks to comparable performance as dense. Our proposed approach provides a more accurate measurement of the training dynamics of sparse networks and can be used to inform future work on the design of networks and optimization techniques that are tailored explicitly to sparsity.

## 2 METHODOLOGY

### 2.1 SAME CAPACITY SPARSE VS DENSE COMPARISON

Our goal is to measure what architecture and optimization choices favor sparse networks relative to dense networks. To fairly compare sparse and dense networks, we propose *Same Capacity Sparse vs Dense Comparison* (`SC-SDC`), a simple framework which allows us to study sparse network optimization and identify what training configurations are not well suited for sparse networks.

`SC-SDC` can be summarized as follows (See Figure 1 for an overview):

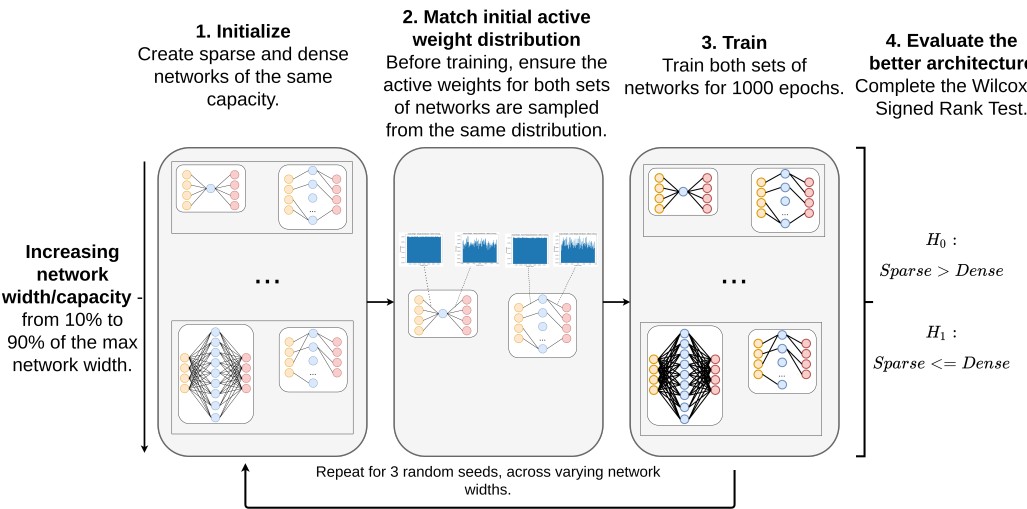

Figure 1: Same Capacity Sparse vs Dense Comparison (SC-SDC)

**1. Initialize** For a chosen network depth (number of layers) $L$ and a maximum network width $N_{MaxW}$, we compare sparse and dense networks at various widths, while ensuring they have the same paramater count.

Initially, we mask the weights $\boldsymbol{\theta}_S$ of sparse network $S$:

$$\boldsymbol{a}_S^l = \boldsymbol{\theta}_S^l \odot m^l \quad , \quad \boldsymbol{a}_D^l = \boldsymbol{\theta}_D^l, \quad \text{for} \quad l = 1, \ldots, L \tag{1}$$

, where $\boldsymbol{\theta}_S^l \odot m^l$ denotes an element-wise product of the weights $\boldsymbol{\theta}_S$ of layer $l$ and the random binary matrix (mask) for layer $l$, $m^l$, $\boldsymbol{a}_S^l$ is the nonzero weights in layer $l$ of sparse network $S$ and $\boldsymbol{a}_D^l$ is the nonzero weights in layer $l$ of dense network $D$ (all the weights since no masking occurs).

For a fair comparison, we need to ensure the same number of nonzero weights for sparse network $S$ and dense network $D$, across each layer $L$.

$$||\boldsymbol{a}_S^l||_0 = ||\boldsymbol{a}_D^l||_0, \quad \text{for} \quad l = 1, \ldots, L \tag{2}$$

We provide more implementation details of how we achieve this in Appendix A.1.

**2. Match active weight distributions** Following prior work (Liu et al., 2018b; Gale et al., 2019), we ensure the nonzero weights at initialization of the sparse and dense networks are sampled from the same distribution at each layer as follows:

$$\boldsymbol{a}_S^l \sim P^l \quad , \quad \boldsymbol{a}_D^l \sim P^l, \quad \text{for} \quad l = 1, \ldots, L \tag{3}$$

, where $P^l$ refers to the initial weight distribution at layer $l$, for example Kaiming initialization (He et al., 2015). This ensures that both sets of active weights (sparse and dense) are initially sampled from the same distribution.

**3. Train** We then train the sparse and dense networks for 1000 epochs (allowing for convergence).

**4. Evaluate the better architecture** We gather the results across the widths/capacity levels and conduct a paired, one-tail Wilcoxon signed-rank test (Wilcoxon, 1945) to evaluate the better architecture. Our null hypothesis ($H_0$) is that sparse networks have similar or worse test accuracy than dense networks (lower or the same median), while our alternative hypothesis ($H_1$) is that sparse networks have better test accuracy performance than dense networks of the same capacity (higher median). This can be formulated as:

$$H_0 : Sparse <= Dense \quad , \quad H_1 : Sparse > Dense \tag{4}$$

Our goal of `SC-SDC` is to compare sparse and dense networks at the same capacity level. By same capacity, we are referring to the same number of active weights, but other notions of same capacity can also be used. We briefly discuss this in Appendix A.1.

Table 1: The average correlation between gradient flow measures and generalization performance

| | Measure | Correlation to Test Loss | | Correlation to Test Accuracy | |
|---|---|---|---|---|---|
| | | Sparse | Dense | Sparse | Dense |
| CIFAR-10 | $\|\boldsymbol{g}\|_1$ (5) | 0.3705 | 0.3940 | 0.3551 | 0.3750 |
| | $\|\boldsymbol{g}\|_2$ (5) | 0.3732 | 0.3181 | 0.3167 | 0.3840 |
| | $egf_1$ (6) | 0.4155 | **0.4168** | **0.3992** | **0.4041** |
| | $egf_2$ (6) | **0.4373** | 0.3323 | 0.3833 | 0.3774 |
| CIFAR-100 | $\|\boldsymbol{g}\|_1$ (5) | 0.4030 | 0.4411 | 0.4286 | 0.3720 |
| | $\|\boldsymbol{g}\|_2$ (5) | 0.3998 | 0.4008 | 0.3974 | 0.3913 |
| | $egf_1$ (6) | **0.4362** | **0.4506** | **0.4418** | 0.3821 |
| | $egf_2$ (6) | 0.4048 | 0.4121 | 0.4142 | **0.3990** |

We compare the average absolute Kendall Rank correlation between different formulations of gradient flow and generalization. The subscript denotes the $p$-norm ($l1$ or $l2$ norm). We see that EGF has higher absolute correlation when compared to standard gradient flow measures. We also see that is consistent across Fashion MNIST (see Appendix B.1).

## 2.2 MEASURING GRADIENT FLOW

Gradient flow (GF) is used to study optimization dynamics and typically approximated by taking the norm of the gradients of the network (Pascanu et al., 2013; Nocedal et al., 2002; Chen et al., 2018; Wang et al., 2020; Evci et al., 2020).

We consider a feedforward neural network $f : \mathbb{R}^D \to \mathbb{R}$, with function inputs $\mathbf{x} \in \mathbb{R}^D$ and network weights $\boldsymbol{\theta}$. The gradient norm is usually computed by concatenating all the gradients of a network into a single vector, $\boldsymbol{g} = \frac{\partial C}{\partial \boldsymbol{\theta}}$ , where $C$ is our cost function. Then the vector norm is taken as follows:

$$gf_p = \|\boldsymbol{g}\|_p, \tag{5}$$

where $p$ denotes the pth-norm.

***Effective Gradient Flow*** Traditional measures of gradient flow take the $l1$ or $l2$ norm of all the gradients (Chen et al., 2018; Pascanu et al., 2013; Evci et al., 2020). This is not appropriate for sparse networks, as this would include gradients of masked weights which have no influence on the forward pass. Furthermore, computing $l1$ or $l2$ across all weights in the networks gives disproportionate influence to layers with more weights. We instead propose a simple modification of Equation 5, which we term *Effective Gradient Flow* (EGF), that computes the average, masked gradient (only gradients of active weights) norm across all layers.

We calculate EGF as follows:

$$\boldsymbol{g} = \left(\frac{\partial C}{\partial \boldsymbol{\theta}^1} \odot m^1, \frac{\partial C}{\partial \boldsymbol{\theta}^2} \odot m^2, \ldots, \frac{\partial C}{\partial \boldsymbol{\theta}^C} \odot m^L\right) \quad \text{for} \quad l = 1, \ldots, L \tag{6}$$

$$egf_p = \frac{\sum_{n=1}^{L} \|\boldsymbol{g}_i\|_p}{L}, \tag{7}$$

where $L$ is number of layers and $\frac{\partial C}{\partial \theta^l} \odot m^l$ denotes an element-wise product of the gradients of layer $l$, $\frac{\partial C}{\partial \theta^l}$ , and the mask $m^l$ applied to the weights of layer $l$. For a fully dense network, $m^l$ is a matrix of all ones, since no gradients are masked.

**EGF has the following favourable properties:**

- **Gradient flow is evenly distributed across layers** EGF distributes the gradient norm across the layers equally, preventing layers with a lot of weights from dominating the measure and also preventing layers with vanishing gradients from being hidden in the formulation, as is the case with equation 5 (when all gradients are appended together).

- **Only gradients of active weights are used** EGF ensures that for sparse networks, only gradients of active weights are used. Even though weights are masked, their gradients are not necessarily

Table 2: Different network configurations for sparse and dense comparisons

| Configuration | Variants |
|---|---|
| Optimizers | SGD , SGD with mom (0.9) , Adagrad , RMSprop and Adam |
| Regularization/Normalization method | No regularization, L2/Weight Decay , Data Augmentation , Skip Connections and Batchnorm. |
| Number of hidden layers | 1 , 2 , 4. |
| Dense Width | 308, 923, 1538, 2153, 2768. |
| Activation functions | ReLU, PReLU , ELU , SReLU and Sigmoid. |
| Batch Size | 128 |
| Learning Rate | 0.001, 0.1 |

zero since the partial derivative of the weight wrt. the loss, is influenced by other weights and activations. Thereby a weight can be zero, but its gradient can be nonzero.

- **Possibility for application in gradient-based pruning methods** Tanaka et al. (2020) showed that gradient-based pruning methods like GRASP (Wang et al., 2020) and SNIP (Lee et al., 2018a), disproportionately prune large layers and are susceptible to layer-collapse, which is when an algorithm prunes all the weights in a specific layer. Due to the fact that `EGF` is evenly distributed across layers, maintaining `EGF` (as opposed to standard gradient norm) could possibly be used as a pruning criteria. Furthermore, current approaches measuring or approximating the change in gradient flow during pruning in sparse networks (Wang et al., 2020; Evci et al., 2020; Singh Lubana & Dick, 2020), could benefit from this new formulation.

To evaluate `EGF` against other standard gradient norm measures, such as the $l1$ and $l2$ norm, we empirically compare these measures and their correlation to test loss and accuracy. We take the absolute average of the Kendall Rank correlation (Kendall, 1938), across the different experiment configurations. We follow a similiar approach to Jiang et al. (2019), but unlike their work which has focused on correlating network complexity measures to the generalization gap, we measure the correlation of gradient flow to performance (accuracy and loss). We measure gradient flow at 10 points evenly spaced throughout training, specifically at the end of epoch 0, 99. 199, 299, 399, 499, 599, 699, 799, 899 and 999.

Our results from Table 1 shows that `EGF` has a higher average absolute correlation to both test loss and accuracy. This is also true of Fashion MNIST (see Appendix B.1). Due to the comparative benefits of `EGF`, we use it for the remainder of the paper to measure the impact of interventions. We include all measures of gradient flow in Appendix B for completeness.

## 2.3 Architecture, Normalization, Regularization and Optimizer Variants

We briefly describe our key experiment variants below, and also include for completeness all unique variants in Table 2.

**Activation functions** ReLU networks (Nair & Hinton, 2010) are known to be more resilient to vanishing gradients than networks that use Sigmoid or Tanh activations, since they only result in vanishing gradients when the input is less than zero, while on active paths, due to ReLU's linearity, the gradients flow uninhibited (Glorot et al., 2011). Although most experiments are run on ReLU networks, we also explore different activation functions, namely PReLU (He et al., 2015), ELU (Clevert et al., 2015), Swish (Ramachandran et al., 2017), SReLU (Jin et al., 2015) and Sigmoid (Neal, 1992).

**Batch normalization and Skip Connections** Other methods to help alleviate the vanishing gradient problem include the addition of skip connections (every two layers) (Srivastava et al., 2015; He et al., 2016) and batch normalization (Ioffe & Szegedy, 2015). We empirically explore these methods.

**Optimization and Regularization techniques** We explore the impact of popular regularization methods: weight decay/$l2$ regularization (0.0001) (Krogh & Hertz, 1992; Hanson & Pratt, 1989) and data augmentation (random crops and random horizontal flipping (Krizhevsky et al., 2012)). Furthermore, we benchmark the impact of the most widely used optimizers such as minibatch stochastic gradient descent (with momentum (0.9) (Sutskever et al., 2013; Polyak, 1964) and without momen-

tum (Robbins & Monro, 1951)) , Adam (Kingma & Ba, 2014), Adagrad (Duchi et al., 2011) and RMSProp (Hinton et al., 2012).

## 3 EMPIRICAL SET-UP

**`SC-SDC` MLP Setting** We use the `SC-SDC` empirical setting (section 2.1) for all experiment variants. We train over 6000 MLPs for 1000 epochs and evaluate performance on CIFAR-10 and CIFAR-100 (Krizhevsky et al., 2009). We compare sparse and dense networks across various widths, depths, learning rates, regularization and optimization methods as shown in Table 2.

We choose a max network width $N_{MaxW}$ of $n + 4$, where $n$ is the input dimension of the network. In the case of CIFAR, $n = 3072$ and so our maximum width $N_{MaxW} = 3076$. We repeat these experiments with one, two and four hidden layers, with the number of active weights in these networks ranging from $949, 256$ to $31, 765, 568$ weights. In section 4, we discuss results achieved using four hidden layers on CIFAR-100 and we provide the one and two hidden layer results in Appendix C.

**Dense Width** Following from `SC-SDC`, these networks are compared at various network widths, specifically a width of $308, 923, 1538, 2153, 2768$ (10%, 30%, 50%, 70% and 90% of our maximum width $N_{MaxW}(3076)$) as shown in Table 2. We use the term **dense width** to refer to the width of a network if that network was dense. For example, when comparing sparse and dense networks at a dense width of 308, this means the dense network has a width of 308, while the sparse network has a width of $N_{MaxW}$ (3076), but has the same number of active connections as the dense counterpart. We provide more detailed discussion of the choices made in the `SC-SDC` implementation in Appendix A.1.

**Extended CNN Setting** We also extend our experiments to Wide Resnet-50 (the WRN-28-10 variant) (Zagoruyko & Komodakis, 2016) and use the optimization and regularization configurations from the paper.

## 4 RESULTS AND DISCUSSION

### 4.1 COMPARISON OF DENSE AND SPARSE INTERVENTIONS USING `SC-SDC`

In this section, we use the results of the Wilcoxon signed rank test from `SC-SDC` to identify where optimization choices are currently well suited for sparse networks and which are not. Furthermore, for each variant, we also measure the gradient flow using $\text{EGF}(egf_2\,(6))$[1] as described in the previous section. Our main findings show that:

1. Batch normalization is critical to training sparse networks, more so than it is for dense networks. This suggests that gradient instability is a key obstacle for sparse optimization.

2. Weight decay (with and without batch normalization) and data augmentation ( without batch normalization) can hurt both sparse and dense network optimization. This particularly occurs when using accelerated, adaptive optimization methods that use an exponentially decaying average of past squared gradients, such as Adam and RMSProp (Ruder, 2016). In these methods, large EGF (gradient flow) strongly correlates to poor test accuracy.

3. Non-saturating activation functions, such as Swish (Ramachandran et al., 2017) and PReLU (He et al., 2015), achieve promising results in both the sparse and dense regime. However, these results are more statistically significant when sparse networks are used and so this could motivate for the use of similar activation functions for training sparse networks.

**Batch normalization plays a disproportionate role in stabilizing sparse networks** Batch normalization ensures that the distribution of the nonlinearity inputs remains stable as the network trains, which was hypothesized to help stabilize gradient propagation (gradients do not explode or vanish) (Ioffe & Szegedy, 2015). Following from Table 3a and 3b, we see that batch normalization is statistically more important for sparse network performance than it is for dense networks, across most configurations and learning rates.

---

[1]Note we measure EGF at 10 points throughout training and take the average

(a) Effect of Different Regularization Methods - 0.001 Learning Rate

|          | NR     | DA     | L2     | BN         | SC     | DA_BN_SC   | DA_L2_BN_SC |
|----------|--------|--------|--------|------------|--------|------------|-------------|
| Adagrad  | 0.9997 | 0.9997 | 0.9983 | **0.0062** | 0.2385 | **0.0006** | **0.0035**  |
| Adam     | **0.0003** | 0.0547 | 0.1984 | 0.0789 | **0.0025** | **0.0391** | 0.1184 |
| RMSProp  | **0.0005** | **0.0003** | 0.3001 | 0.1165 | 0.1664 | 0.0957 | **0.0234** |
| SGD      | 0.9997 | 0.9997 | 0.9997 | **0.0003** | 0.2478 | **0.0013** | **0.0035** |
| Mom (0.9)| 0.9997 | 0.9997 | 0.9995 | **0.0006** | 0.9989 | **0.0005** | **0.0025** |

(b) Effect of Different Regularization Methods - 0.1 Learning Rate

|          | BN         | BN_SC      | DA_BN      | L2_BN  | DA_BN_SC   | DA_L2_BN_SC |
|----------|------------|------------|------------|--------|------------|-------------|
| Adagrad  | **0.0003** | **0.0016** | **0.0018** | 0.9626 | **0.0234** | **0.0140**  |
| Adam     | 0.0698     | **0.0035** | **0.0023** | **0.0434** | 0.1912 | 0.3766  |
| RMSProp  | **0.0021** | 0.5624     | **0.0267** | **0.0079** | 0.8943 | **0.0016** |
| SGD      | **0.0005** | **0.0045** | **0.0003** | **0.0481** | **0.0004** | **0.0126** |
| Mom (0.9)| **0.0004** | **0.0027** | **0.0016** | 0.4435 | **0.0053** | 0.2118  |

(c) Effect of Different Activation Functions - 0.1 Learning Rate

|          | Relu       | Swish      | PRelu      | SRelu  | Sigmoid    | Elu        |
|----------|------------|------------|------------|--------|------------|------------|
| Adagrad  | **0.0234** | **0.0045** | **0.0498** | 0.1817 | 0.4797     | **0.0032** |
| Adam     | 0.1912     | 0.1817     | **0.0391** | 0.0620 | **0.0045** | **0.0004** |
| RMSProp  | 0.8943     | 0.1671     | **0.0023** | **0.0115** | 0.9973 | 0.1533     |
| SGD      | **0.0126** | **0.0267** | **0.0045** | 0.0778 | **0.0304** | 0.0559     |
| Mom (0.9)| 0.2118     | **0.0134** | **0.0013** | 0.0778 | **0.0013** | 0.9733     |

*NR - No Regularization, BN - Batchnorm, SC - Skip Connections, DA - Data Augmention and L2- weight decay.*

Table 3: Wilcoxon Signed Rank Test Results for ReLU networks with four hidden layers, trained on CIFAR-100, using different learning rates. We use a $p$-value of 0.05, the bold values indicate where sparse networks perform better than dense networks in a statistical significance manner (reject $H_0$ from 4), while non-bold values indicate that it is possible dense networks have the same or better test accuracy in that configuration. The performance results for these networks are presented in Figure 2a, 12 and 10a .

From Table 3a and Figure 10a, we see methods such as L2 and data augmentation usually favour dense networks (apart from Adam and RMSProp, when using data augmentation). However, with the addition of batch normalization (Table 3b, L2 to L2_BN and DA to DA_BN ), these methods favour sparse variants. This is especially apparent in Figure 10a, where batch normalization improves performance across all sparse optimizers, while resulting in a lower, more stable EGF. This further emphasizes the importance of stabilizing gradient flow, particularly in sparse networks.

**Weight Decay and Data Augmentation can hurt sparse network optimization** When we take a closer look at the effects of weight decay and data augmentation on sparse network accuracy (Figure 2,Figure 13), we see that weight decay (even with batch normalization) drastically decreases accuracy when used with adaptive optimization methods that use an exponentially decaying average of past squared gradients (Adam and RMSProp). Furthermore, it results in distinctively larger EGF values, which hints at Adam and RMSProp being more sensitive to larger gradient norms than other optimizers. This agrees with Loshchilov & Hutter (2017), who proposed a different formulation of weight decay for adaptive methods, since the current $L2$ regularization formulation for adaptive methods could lead to weights with large gradients being regularized less, although this was not experimentally verified. In the context of data augmentation, we see poor test accuracy when it is used without batch normalization (Figure 10a). If used with batch normalization (Figure 2), it results in a lower EGF and best test accuracy. This further emphasized the need to stabilize gradient flow and how EGF can be used to this end.

**The potential of non-saturating activation functions - Swish and PReLU** We also explore the effect of different activation functions on sparse network optimization. For the activation function

(a) Test Accuracy for Dense and Sparse Networks on CIFAR-100

(b) Gradient Flow for Dense and Sparse Networks on CIFAR-100

*NR - No Regularization, BN - Batchnorm, SC - Skip Connections, DA - Data Augmention and L2- weight decay.*

Figure 2: We show the test accuracy (upper image) and gradient flow (lower image) results for Sparse MLPs with four hidden layers and a large learning rate (0.1), across different regularization methods and promising activations. The results for all optimizers can be found in Figure 13.

variants, the best configuration for each optimizer was chosen. For Adagrad, Adam and RMSProp we use BN, SC and DA, while for SGD, we use BN, SC, L2 and DA.

From Table 3c, we see that Swish, PReLU and Sigmoid favour sparse architectures, but from the performance results from Figure 12, we see that only Swish and PReLU are viable activation choices. We continue to see a consistent trend for adaptive methods (most notably in Adam and RMSProp), that higher EGF values, for example in SReLU, correspond to poor performance (Figure 11b), while promising methods result in a lower EGF value (such as Swish). This further emphasizes how EGF can be used to guide advances in network optimization.

## 4.2 GENERALIZATION OF RESULTS ACROSS ARCHITECTURE TYPES.

In this section, we move on from `SC-SDC` and extend our result to Wide ResNet-50. We note from Figure 3, that most of our results from SC-SDC also hold on larger, more complicated models. We see that $L2$ regularization (even with batch normalization) hurts performance for adaptive methods (Adagrad and Adam) and also results in higher `EGF` values (Figure 14). Furthermore, we also see data augmentation is beneficial when used with batch normalization. Finally, we see that Swish is a promising activation function for adaptive methods and leads to lower `EGF` (Figure 14). This shows that the `SC-SDC` results are not constrained to small scale experiments and that it can be used to learn about dynamics of larger, more complicated networks.

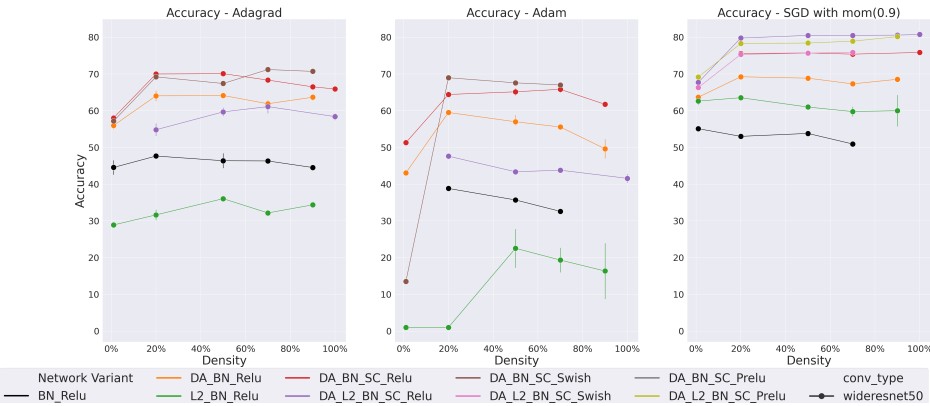

Figure 3: WideResNet50 Test Accuracy on CIFAR-100. The density ranges from 1% to 100%. The gradient flow results can be found in Figure 14.

## 5 RELATED WORK

**Pruning at Initialization** Methods that prune at initialization aim to start sparse, instead of first pre-training an overparameterized network and then pruning. These methods use certain criteria to estimate at initialization, which weights should remain active. This criteria includes using the connection sensitivity (Lee et al., 2018b), gradient flow (via the Hessian vector product) (Wang et al., 2020) and conversation of synaptic saliency (Tanaka et al., 2020). Another branch of pruning is Dynamic Sparse Training, which uses information gathered during the training process, to dynamically update the sparsity pattern of these sparse networks (Mostafa & Wang, 2019; Bellec et al., 2017; Mocanu et al., 2018; Dettmers & Zettlemoyer, 2019; Evci et al., 2019). While our work is motivated by the same goal of allowing networks to start sparse and converge to the same performance as dense networks, we instead focus on the impact of optimization and regularization choices on sparse networks.

**Sparse Network Optimization as Pruning Criteria** Optimization in sparse networks has often been neglected in favour of studying network initialization. However, there has been work that has looked at sparse network optimization from different perspectives, mainly as a guide for pruning criteria. This includes using gradient information (Mozer & Smolensky, 1989; LeCun et al., 1989; Hassibi & Stork, 1992; Karnin, 1990), approximates of gradient flow (Wang et al., 2020; Dettmers & Zettlemoyer, 2019; Evci et al., 2020) and Neural Tangent Kernel (NTK) (Liu & Zenke, 2020) to guide the introduction of sparsity.

**Sparse Network Optimization to study Network Dynamics** Apart from use as pruning criteria, optimization information has been used to investigate aspects of sparse networks, such as their loss landscape (Evci et al., 2019), how they are impacted by SGD noise (Frankle et al., 2019a), the effect of different activation functions (Dubowski, 2020) and their weight initialization (Lee et al., 2019). Our work differs from these approaches as we consider more aspects of the optimization and regularization process in a controlled experimental setting (SC-SDC), while using `EGF` to reason about some of the results.

## 6 CONCLUSION AND FUTURE WORK

In this work, we take a wider view of sparse optimization strategies and introduce appropriate tooling to measure the impact of architecture and optimization choices on sparse networks (`EGF` , SC-SDC ). Our results show that weight decay and data augmentation can hurt optimization, when adaptive optimization methods are used and this usually corresponds to a much higher `EGF`.Furthermore, we show how batch normalization is critical to training sparse networks, more so than it is for dense networks as it helps stabilize gradient flow. We also show the potential of non-saturating activation functions for sparse networks such as Swish and PReLU. Finally, we show that our results extend to more complicated models like Wide ResNet-50.

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

# A    SC-SDC

In this section, we provide more information about SC-SDC and its benefits.

## A.1    SC-SDC IMPLEMENTATION DETAILS

**Wilcoxon Signed Rank Test** This is a non-parametric test that compares dependent or paired samples, without assuming the differences in between the paired experiments are normally distributed (McDonald, 2009; Demšar, 2006).

**Random Sparsity** Our work focuses on the training dynamics of random, sparse networks. This ensures that what is learned is not dependent on a specific pruning method, but rather can be used to better understand sparse training in general. Going forward, it would be interesting to explore these dynamics on pruned networks.

We achieve random sparsity, by generating a random mask for each layer and then multiply the weights by this mask during each forward pass. The sparsity is distributed evenly across the network. For example, a 20% sparse MLP has 20% of the weights remaining in each layer.

**Dense Width** A critical component to how we specify our experiments is a term we define as dense width. In order to fairly compare sparse and dense networks, we need them to have the same number of active connections at each depth. In the case of sparse networks, this means ensuring they have the same number of active connections as the dense networks, while remaining sparse. Dense width refers to the width of a network if that network was dense. This process of comparing sparse and dense networks at different dense widths is illustrated in figure 5.

**Fair comparison of Sparse and Dense networks** As can be seen from figure 5, SC-SDC ensures the exact same active parameter count, but the sparse networks will be connected to more neurons. It is possible that the increased number of activations being used can lead to sparse networks having higher representational power, however most work on expressivity of neural networks looks at this from a depth perspective and proves certain depths of networks are universal approximators (Eldan & Shamir, 2016; Hornik et al., 1989; Funahashi, 1989).

To this end, we ensure these networks have the same depth, but we believe going forward an interesting direction would be ensuring they have a similar amount of active neurons.

**SC-SDC comparison details** For completeness, we provide more details of how we ensure sparse and dense networks are of the same capacity.

Following from equation 2, to ensure the same number of weights in sparse and dense networks, we can ensure they have the same number of active weights at each layer as follows:

$$||\boldsymbol{a}_S^l||_0 = ||\boldsymbol{a}_D^l||_0, \quad \text{for} \quad l = 1, \ldots, L \tag{8}$$

This is achieved by masking each of the weight layers of sparse network $S$:

$$\boldsymbol{a}_S^l = \boldsymbol{\theta}_S^l \odot m^l \quad \text{for} \quad l = 1, \ldots, L \tag{9}$$

, where $m^l$ is a random binary matrix (mask) for layer $l$, s.t. $||m^l||_0 = \boldsymbol{a}_D^l$, where $\boldsymbol{a}_D^l$ is determined by the chosen capacity, these networks will be compared at.

For SC-SDC, we need a maximum network width $N_{MaxW}$ and comparison width $N_W$. We choose a max network width $N_{MaxW}$ of $n + 4$, where $n$ is the input dimension of the network. In the case of CIFAR, $n = 3072$ and so our maximum width $N_{MaxW} = 3076$. The choice of $n + 4$ follows from Lu et al. (2017), where the authors prove a universal approximation theorem for width-bounded ReLU networks, with width bounded to $n + 4$. Our comparison width, $N_W$, is equivalent to dense widths we vary in our experiments - 308, 923, 1538, 2153, 2768.

The dimensions of each of layers are as follows:

1. First Layer:

$$\boldsymbol{\theta}_D^1 \in \mathbb{R}^{I \times N_W} \quad , \quad \boldsymbol{\theta}_S^1 \in \mathbb{R}^{I \times N_{MaxW}} \quad , \quad m^1 \in \{0,1\}^{I \times N_{MaxW}} \tag{10}$$

2. Intermediate Layers:

$$\boldsymbol{\theta}_S^1 \in \mathbb{R}^{N_W \times N_W} \quad , \quad \boldsymbol{\theta}_S^1 \in \mathbb{R}^{N_{MaxW} \times N_{MaxW}} \quad , \quad m^{\{2,\dots,L-1\}} \in \{0,1\}^{N_{MaxW} \times N_{MaxW}}$$
(11)

3. Final Layer:

$$\boldsymbol{\theta}_S^L \in \mathbb{R}^{N_W \times O} \quad , \quad \boldsymbol{\theta}_S^L \in \mathbb{R}^{N_{MaxW} \times O} \quad , \quad m^L \in \{0,1\}^{N_{MaxW} \times O}$$
(12)

, where $N_{MaxW}$ is maximum width of the sparse layer, $N_W$ is the comparison width, $I$ is the input dimension, $O$ is output dimension, $L$ is the number of layers in the network, $\boldsymbol{\theta}_S^l$ is the weights in layer $l$ of sparse network $S$ and $\boldsymbol{\theta}_D^l$ is the weights in layer $l$ of dense network $D$.

This process would be the same for convolutional layers, but there would be a third dimension to handle the different channels. In figure 4, we provide an illustrative example showing how to ensure sparse and dense networks are compared fairly.

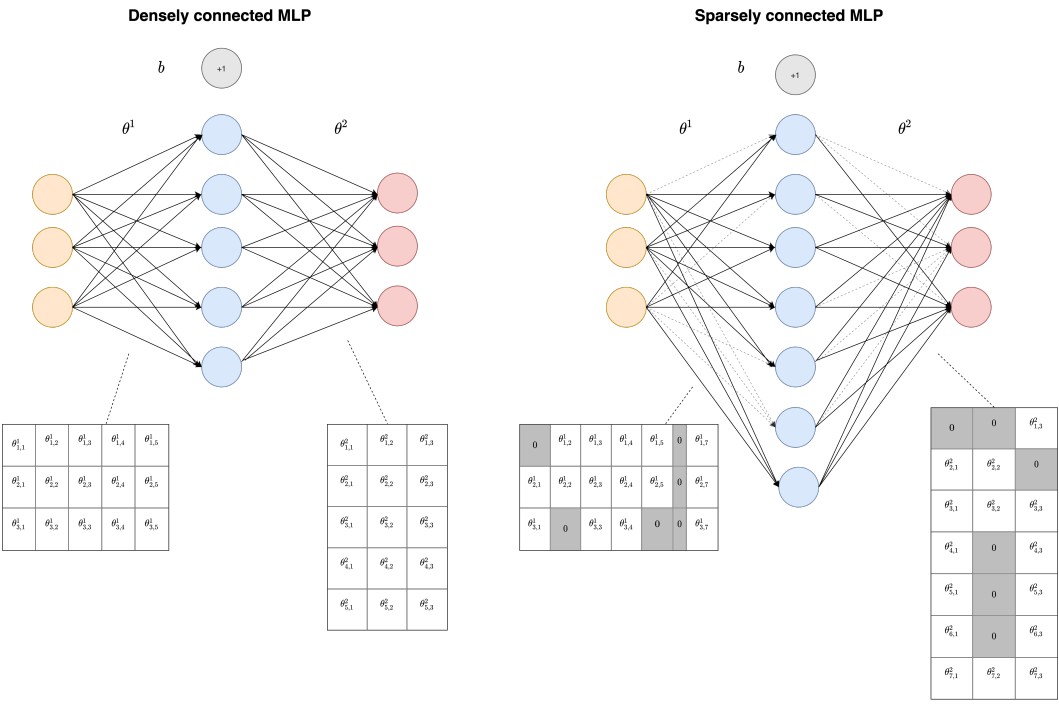

Figure 4: Fair comparison of sparse and dense neural networks

## A.2 BENEFITS

The benefits of `SC-SDC` can be summarized as follows:

- **We can better understand sparse network optimization.** `SC-SDC` allows us to identify which optimization or regularization methods are poorly suited to sparse networks in a controlled setting, ensuring the results are a direct result of the sparse connections themselves.

- **Learn at what parameter and size budget, sparse networks are better than dense.** Comparing sparse and dense networks of the same capacity allows us to see which architecture is better at different configurations. In configurations where sparse architectures perform better, we could exploit advances in sparse matrix computation and storage (Zhao et al., 2018; Merrill & Garland, 2016) to simply default to sparse architectures.

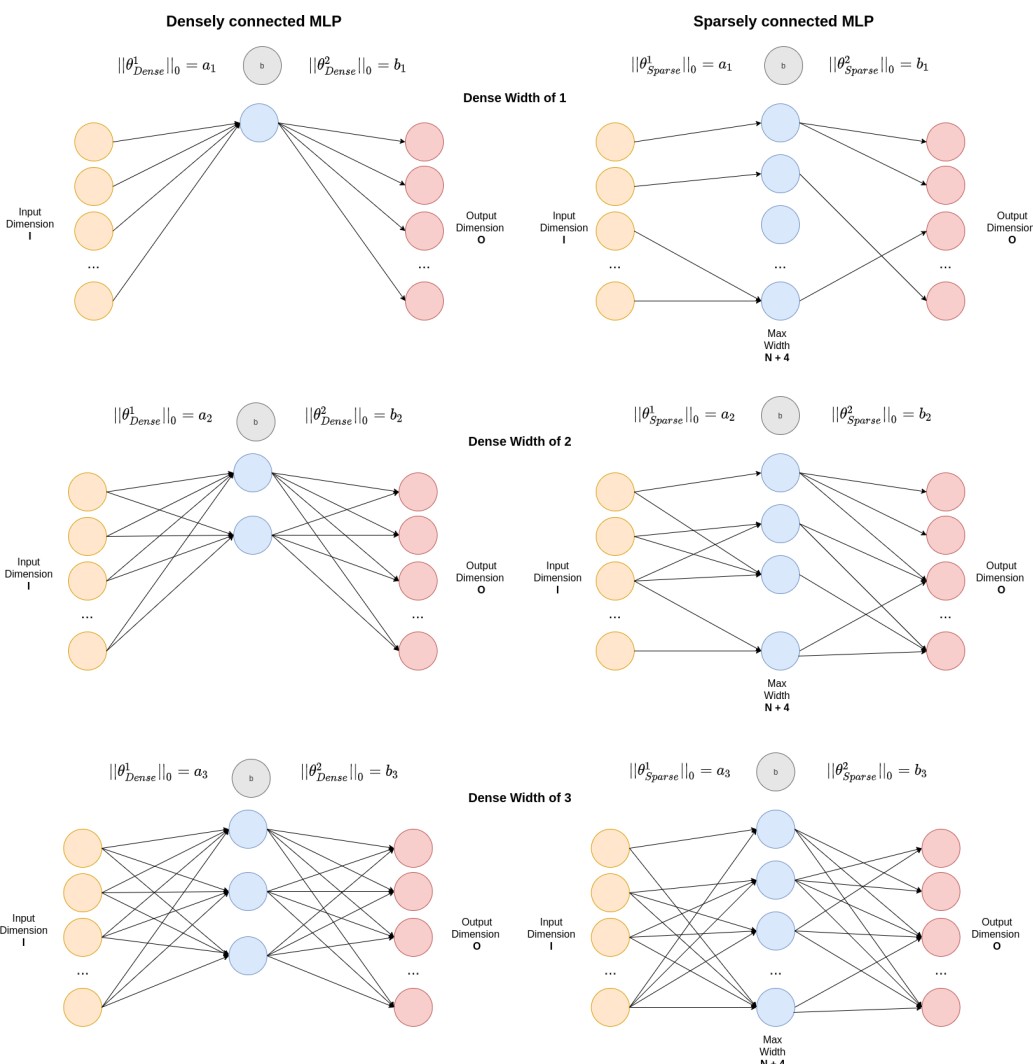

Figure 5: Comparing sparse and dense neural network fairly at different widths

# B  GRADIENT FLOW

## B.1  EGF RESULTS ON FMNIST

We extend our experiments to Fashion MNIST (Xiao et al., 2017), a dataset that is distinctively different to the CIFAR datasets we used in section 2.2. We ran 450 experiments with networks with four hidden layers, using 0.001 as a learning rate and for 500 epochs. We varied configurations as follows:

- Optimizers - Adagrad, Adam and SGD with momentum.
- Regularization methods - no regularization, batchnorm, skip connections, l2 (0.0001) and data augmentation.

From table 4, we see that out of the gradient flow formulations, EGF still correlates better to generalization performance.

For completeness, we present the full set of results using the different formulations of gradient flow on CIFAR-100. Namely, we show $||\boldsymbol{g}||_1$ (5) (Figure 6) ,$||\boldsymbol{g}||_2$ (5) (Figure 7) ,$egf_1$ (6) (Figure 8) and $egf_2$ (6) (Figure 9).

Table 4: The average correlation between gradient flow measures and generalization performance for FMIST

| | Measure | Correlation to Test Loss | | Correlation to Test Accuracy | |
|---|---|---|---|---|---|
| | | Sparse | Dense | Sparse | Dense |
| FMNIST | $\|\boldsymbol{g}\|_1$ (5) | 0.3259 | 0.2522 | 0.3536 | **0.3487** |
| | $\|\boldsymbol{g}\|_2$ (5) | 0.3207 | 0.2702 | 0.3139 | 0.3318 |
| | $egf_1$ (6) | 0.3534 | 0.2522 | **0.3748** | **0.3487** |
| | $egf_2$ (6) | **0.3672** | **0.3017** | 0.2314 | 0.3335 |

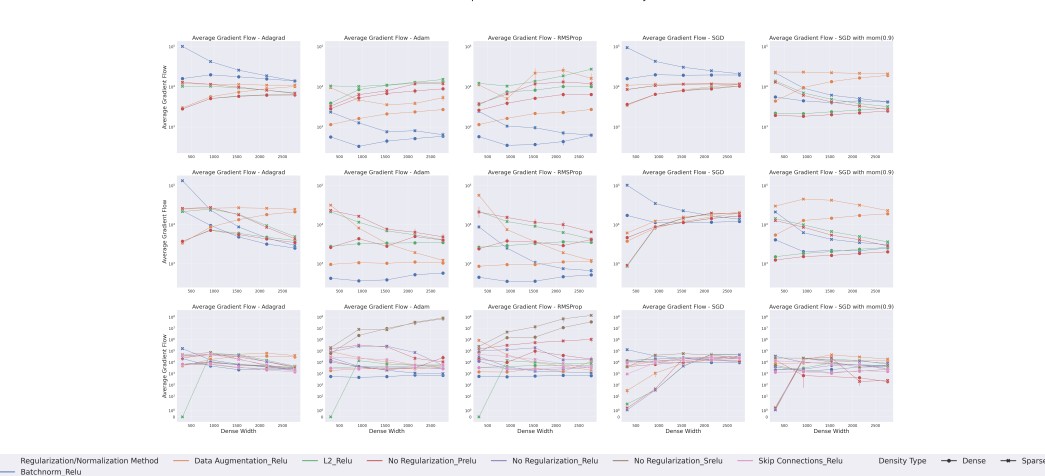

Figure 6: Gradient Flow in CIFAR-100 using $\|\boldsymbol{g}\|_1$

# C  DETAILED RESULTS FOR SC-SDC

In this section, we presented the detailed results for our experiments.

1. Detailed Results with a low learning rate (0.001).
2. Detailed Results with a high learning rate (0.1).
3. Results for different activation functions.

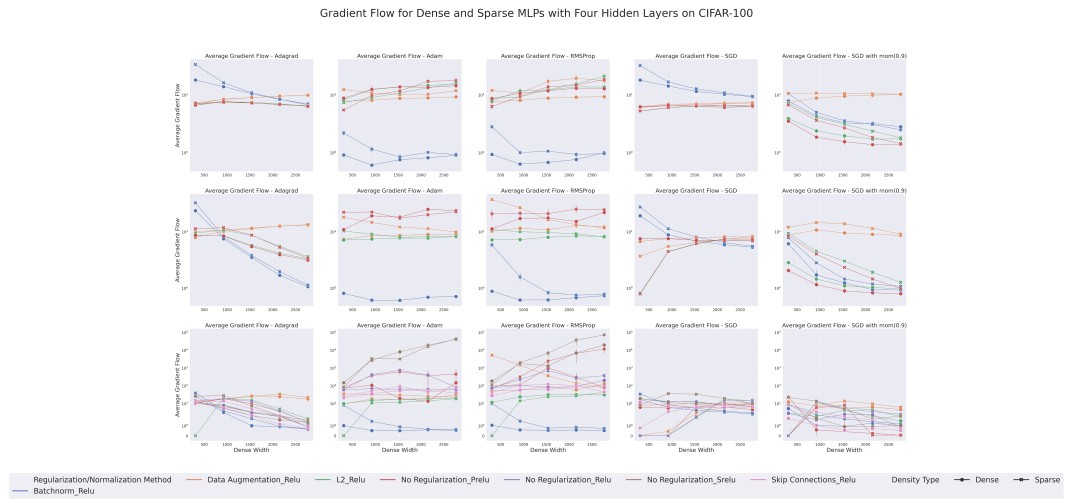

Figure 7: Gradient Flow in CIFAR-100 using $||g||_2$

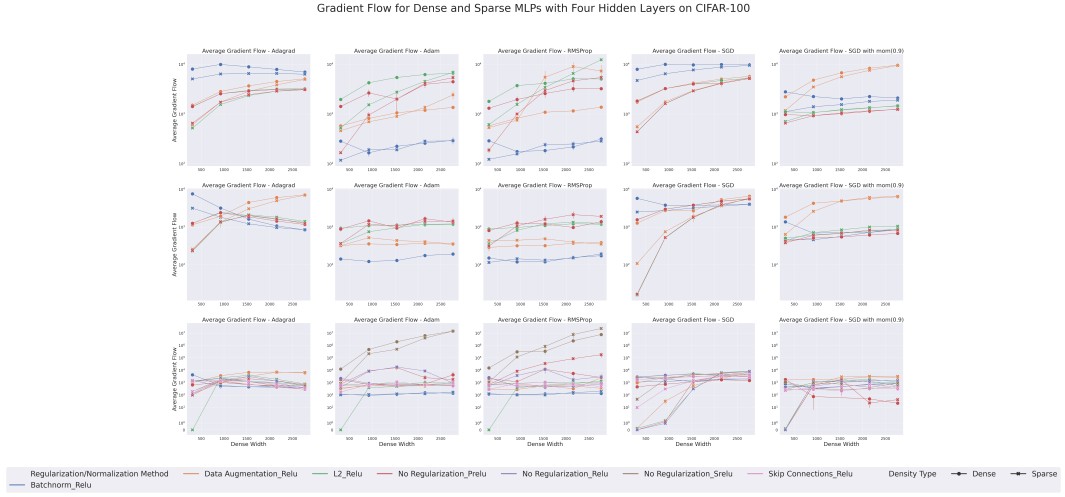

Figure 8: Gradient Flow in CIFAR-100 using $egf_1$

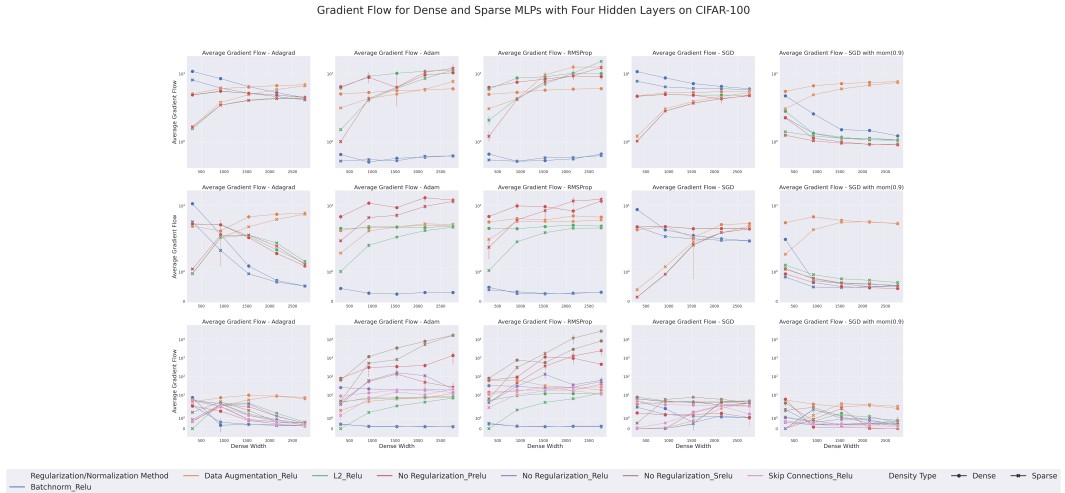

Figure 9: Gradient Flow in CIFAR-100 using $egf_2$

Figure 10: Effect of Regularization on Accuracy and Gradient Flow for Dense and Sparse Networks on CIFAR-100, with low learning rate (0.001)

(a) Test Accuracy for Dense and Sparse Networks on CIFAR-100

(b) Gradient Flow for Dense and Sparse Networks on CIFAR-100

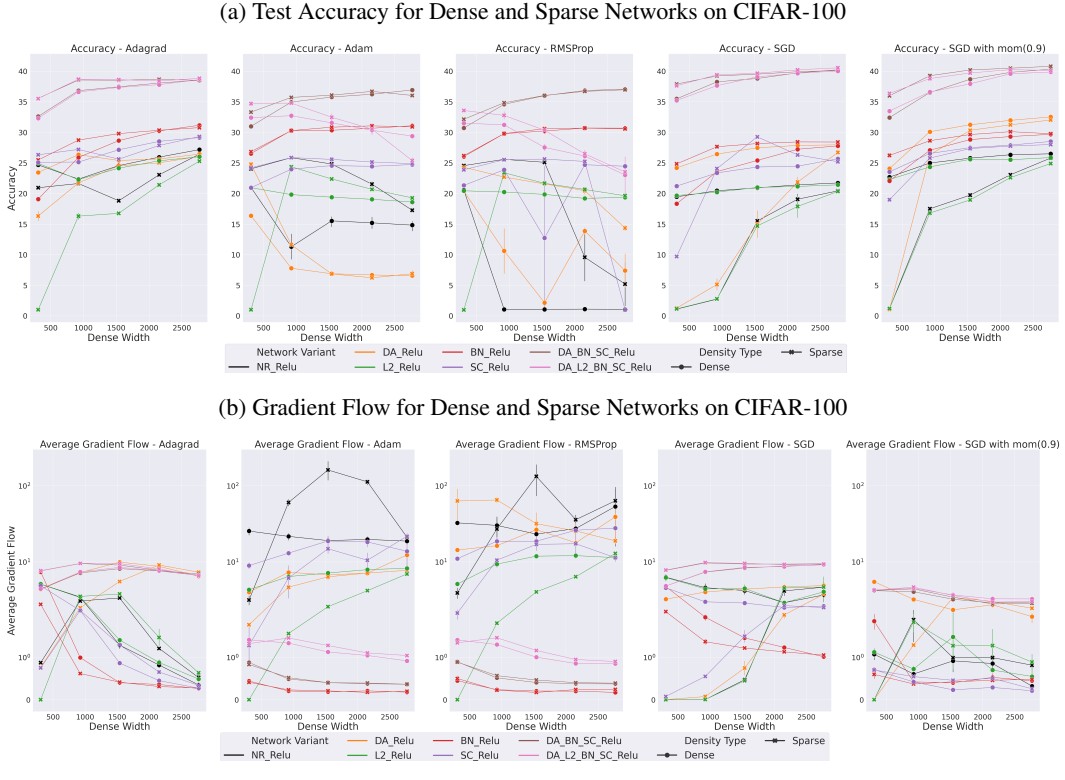

Figure 11: Effect of Activation Functions on Accuracy and Gradient Flow for Dense and Sparse Networks on CIFAR-100, with large learning rate (0.1)

(a) Test Accuracy for Dense and Sparse Networks on CIFAR-100

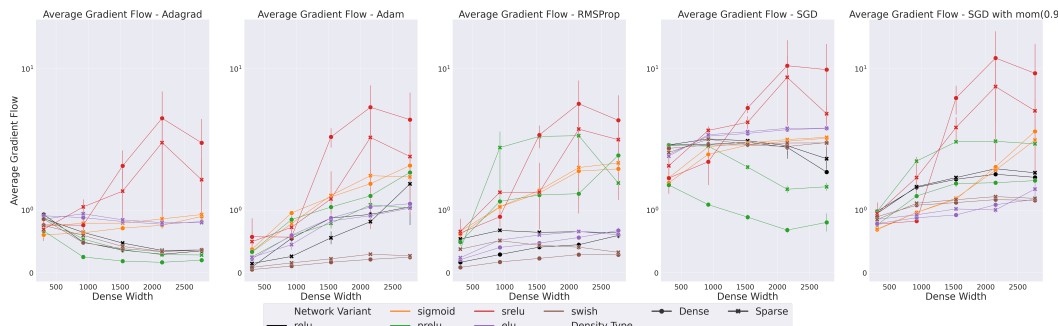

(b) Gradient Flow for Dense and Sparse Networks on CIFAR-100

Figure 12: Test Accuracy for Dense and Sparse Networks on CIFAR-100. Each configuration is the best configuration for that optimizer. For Adagrad, Adam and RMSProp we use BN, SC and DA, while for SGD, we use BN, SC, L2 and DA.

Table 5: Test Accuracy summary for CIFAR-10 with low learning rate (0.001)

(a) One Hidden Layer

| | No Regularization | | Data Augmentation | | L2 | | Batchnorm | |
|---|---|---|---|---|---|---|---|---|
| | Dense | Sparse | Dense | Sparse | Dense | Sparse | Dense | Sparse |
| Adagrad | 54.537 +/- 0.91 | 55.259 +/- 0.36 | 56.098 +/- 2.395 | 55.497 +/- 3.022 | 54.565 +/- 0.909 | 55.269 +/- 0.545 | 54.279 +/- 2.034 | 55.981 +/- 0.349 |
| Adam | 51.391 +/- 1.066 | 52.476 +/- 0.48 | 37.955 +/- 2.874 | 42.404 +/- 6.494 | 48.671 +/- 0.766 | 49.312 +/- 1.033 | 52.668 +/- 1.275 | 53.321 +/- 0.892 |
| RMSProp | 48.999 +/- 0.76 | 51.507 +/- 0.817 | 36.289 +/- 4.053 | 40.62 +/- 7.105 | 47.7 +/- 1.926 | 48.145 +/- 2.468 | 52.581 +/- 1.617 | 53.045 +/- 1.282 |
| SGD | 53.714 +/- 1.564 | 55.023 +/- 0.48 | 57.091 +/- 1.524 | 55.089 +/- 3.08 | 53.775 +/- 1.559 | 55.091 +/- 0.46 | 54.364 +/- 2.363 | 55.996 +/- 0.696 |
| SGD with mom(0.9) | 53.774 +/- 1.968 | 54.746 +/- 0.804 | 56.98 +/- 1.027 | 57.849 +/- 0.754 | 54.083 +/- 1.72 | 55.051 +/- 1.06 | 54.684 +/- 2.083 | 55.951 +/- 0.604 |

(b) Two Hidden Layers

| | No Regularization | | Data Augmentation | | L2 | | Batchnorm | |
|---|---|---|---|---|---|---|---|---|
| | Dense | Sparse | Dense | Sparse | Dense | Sparse | Dense | Sparse |
| Adagrad | 54.179 +/- 1.92 | 55.333 +/- 0.558 | 59.739 +/- 3.03 | 58.967 +/- 4.313 | 53.813 +/- 1.783 | 55.123 +/- 0.301 | 53.703 +/- 2.711 | 56.471 +/- 0.317 |
| Adam | 52.003 +/- 0.91 | 53.785 +/- 1.146 | 42.095 +/- 7.723 | 49.468 +/- 7.948 | 50.588 +/- 0.683 | 51.681 +/- 0.668 | 57.541 +/- 1.525 | 57.886 +/- 1.171 |
| RMSProp | 51.352 +/- 1.198 | 53.499 +/- 1.5 | 46.872 +/- 4.916 | 51.919 +/- 5.436 | 50.382 +/- 0.602 | 50.52 +/- 0.918 | 57.432 +/- 1.512 | 57.974 +/- 1.151 |
| SGD | 52.895 +/- 1.918 | 53.905 +/- 1.735 | 59.893 +/- 2.16 | 55.865 +/- 5.475 | 52.931 +/- 1.852 | 53.941 +/- 1.724 | 54.153 +/- 2.259 | 56.459 +/- 0.413 |
| SGD with mom(0.9) | 53.777 +/- 1.583 | 53.357 +/- 3.18 | 61.814 +/- 1.959 | 62.367 +/- 2.177 | 53.835 +/- 1.728 | 53.26 +/- 3.309 | 56.081 +/- 2.158 | 57.575 +/- 0.428 |

(c) Four Hidden Layers

| | No Regularization | | Data Augmentation | | L2 | | Batchnorm | | Skip Connections | |
|---|---|---|---|---|---|---|---|---|---|---|
| | Dense | Sparse | Dense | Sparse | Dense | Sparse | Dense | Sparse | Dense | Sparse |
| Adagrad | 53.933 +/- 3.137 | 53.837 +/- 2.903 | 58.118 +/- 2.059 | 56.836 +/- 5.211 | 53.755 +/- 3.108 | 44.738 +/- 18.207 | 55.204 +/- 3.483 | 57.153 +/- 1.194 | 53.896 +/- 3.159 | 55.491 +/- 1.362 |
| Adam | 24.538 +/- 17.34 | 36.206 +/- 22.015 | 46.911 +/- 5.617 | 49.115 +/- 6.169 | 50.875 +/- 0.63 | 43.025 +/- 17.121 | 58.379 +/- 1.538 | 58.947 +/- 1.261 | 52.981 +/- 0.918 | 54.54 +/- 0.895 |
| RMSProp | 24.241 +/- 15.653 | 45.916 +/- 12.861 | 53.122 +/- 3.217 | 54.358 +/- 3.115 | 50.249 +/- 0.631 | 41.931 +/- 17.325 | 58.101 +/- 1.318 | 58.665 +/- 1.248 | 52.356 +/- 1.692 | 53.987 +/- 0.756 |
| SGD | 50.469 +/- 1.471 | 36.697 +/- 16.941 | 60.295 +/- 1.784 | 46.193 +/- 18.397 | 50.389 +/- 1.562 | 36.478 +/- 16.985 | 52.78 +/- 2.313 | 55.299 +/- 0.347 | 51.949 +/- 1.698 | 53.941 +/- 1.082 |
| SGD with mom(0.9) | 54.749 +/- 1.473 | 45.291 +/- 14.629 | 61.43 +/- 2.792 | 53.408 +/- 19.158 | 54.382 +/- 1.506 | 44.661 +/- 14.911 | 56.496 +/- 2.326 | 58.336 +/- 0.632 | 54.298 +/- 1.738 | 53.312 +/- 3.529 |

Figure 13: Effect of Different Interventions on Accuracy and Gradient Flow for Dense and Sparse Networks on CIFAR-100, with large learning rate (0.1) - All Optims

(a) Test Accuracy for Dense and Sparse Networks on CIFAR-100

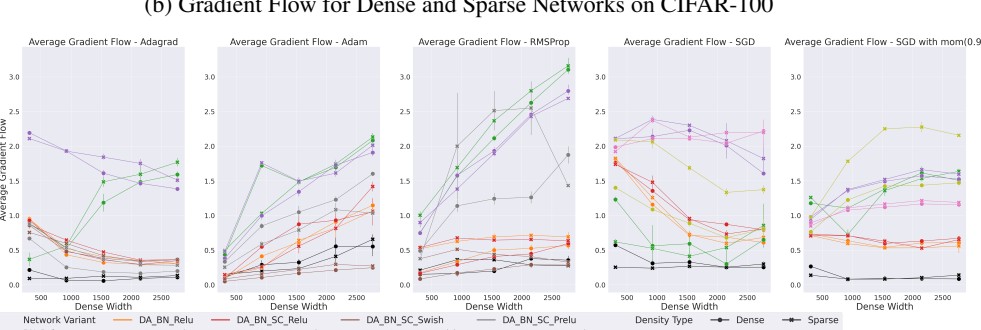

(b) Gradient Flow for Dense and Sparse Networks on CIFAR-100

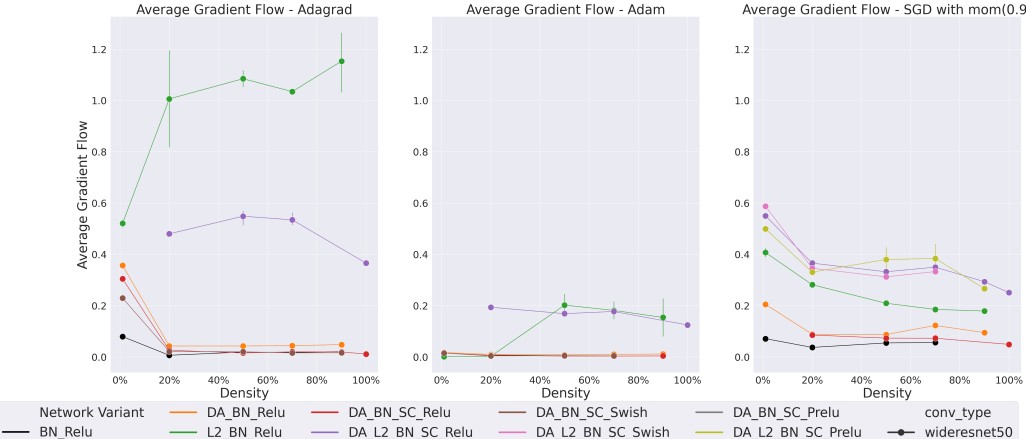

Figure 14: WideResNet50 Grad Flow on CIFAR-100. The density ranges from 1% to 100%.

Table 6: Test Loss summary for CIFAR-10 with low learning rate (0.001)

(a) One Hidden Layer

| | No Regularization | | Data Augmentation | | L2 | | Batchnorm | |
|---|---|---|---|---|---|---|---|---|
| | Dense | Sparse | Dense | Sparse | Dense | Sparse | Dense | Sparse |
| Adagrad | 1.768 +/- 0.156 | 1.677 +/- 0.216 | 1.274 +/- 0.066 | 1.29 +/- 0.084 | 1.712 +/- 0.127 | 1.615 +/- 0.18 | 1.564 +/- 0.033 | 1.548 +/- 0.079 |
| Adam | 142.604 +/- 46.942 | 102.403 +/- 61.214 | 11.263 +/- 6.312 | 9.691 +/- 7.296 | 6.296 +/- 1.944 | 5.542 +/- 1.923 | 6.242 +/- 0.969 | 6.144 +/- 0.972 |
| RMSProp | 70.0 +/- 15.672 | 62.098 +/- 25.008 | 12.061 +/- 8.758 | 25.578 +/- 26.538 | 4.582 +/- 1.216 | 5.503 +/- 1.938 | 6.644 +/- 1.002 | 7.31 +/- 1.16 |
| SGD | 1.977 +/- 0.168 | 1.616 +/- 0.161 | 1.253 +/- 0.04 | 1.306 +/- 0.084 | 1.939 +/- 0.166 | 1.595 +/- 0.151 | 1.699 +/- 0.069 | 1.663 +/- 0.017 |
| SGD with mom(0.9) | 3.076 +/- 0.56 | 2.598 +/- 0.047 | 1.438 +/- 0.112 | 1.407 +/- 0.113 | 2.42 +/- 0.413 | 2.092 +/- 0.055 | 2.208 +/- 0.122 | 2.145 +/- 0.021 |

(b) Two Hidden Layers

| | No Regularization | | Data Augmentation | | L2 | | Batchnorm | |
|---|---|---|---|---|---|---|---|---|
| | Dense | Sparse | Dense | Sparse | Dense | Sparse | Dense | Sparse |
| Adagrad | 2.962 +/- 0.329 | 2.53 +/- 0.679 | 1.178 +/- 0.067 | 1.198 +/- 0.107 | 2.621 +/- 0.2 | 2.151 +/- 0.442 | 2.62 +/- 0.089 | 2.161 +/- 0.339 |
| Adam | 116.26 +/- 25.299 | 143.085 +/- 69.247 | 2.006 +/- 0.305 | 2.232 +/- 0.474 | 3.698 +/- 0.234 | 3.379 +/- 0.4 | 7.469 +/- 1.768 | 7.309 +/- 1.877 |
| RMSProp | 148.115 +/- 30.097 | 161.511 +/- 74.983 | 2.192 +/- 0.179 | 2.832 +/- 0.981 | 4.339 +/- 0.144 | 3.985 +/- 0.514 | 7.416 +/- 1.368 | 7.193 +/- 1.495 |
| SGD | 2.917 +/- 0.488 | 2.008 +/- 0.366 | 1.157 +/- 0.051 | 1.268 +/- 0.145 | 2.809 +/- 0.47 | 1.959 +/- 0.336 | 2.41 +/- 0.249 | 2.01 +/- 0.134 |
| SGD with mom(0.9) | 4.054 +/- 0.818 | 3.971 +/- 1.438 | 2.442 +/- 0.452 | 1.79 +/- 0.503 | 2.95 +/- 0.574 | 2.948 +/- 1.025 | 2.788 +/- 0.359 | 2.376 +/- 0.128 |

(c) Four Hidden Layers

| | No Regularization | | Data Augmentation | | L2 | | Batchnorm | | Skip Connections | |
|---|---|---|---|---|---|---|---|---|---|---|
| | Dense | Sparse | Dense | Sparse | Dense | Sparse | Dense | Sparse | Dense | Sparse |
| Adagrad | 5.158 +/- 0.637 | 4.361 +/- 1.604 | 2.021 +/- 0.752 | 1.838 +/- 0.697 | 3.819 +/- 1.308 | 3.172 +/- 0.743 | 2.802 +/- 0.29 | 2.396 +/- 0.225 | 4.897 +/- 0.682 | 3.625 +/- 1.131 |
| Adam | 11.182 +/- 11.38 | 95.498 +/- 157.421 | 2.722 +/- 0.382 | 2.816 +/- 0.681 | 3.608 +/- 0.171 | 3.51 +/- 0.671 | 7.204 +/- 0.9 | 7.181 +/- 1.162 | 244.704 +/- 90.922 | 247.345 +/- 134.383 |
| RMSProp | 12.352 +/- 9.63 | 191.175 +/- 318.66 | 5.452 +/- 2.659 | 6.504 +/- 4.251 | 3.647 +/- 0.322 | 3.718 +/- 0.857 | 7.645 +/- 0.867 | 7.623 +/- 1.159 | 194.814 +/- 77.03 | 199.102 +/- 106.311 |
| SGD | 5.139 +/- 0.855 | 3.578 +/- 1.253 | 1.173 +/- 0.033 | 1.518 +/- 0.472 | 4.812 +/- 0.829 | 3.458 +/- 1.102 | 2.779 +/- 0.365 | 2.243 +/- 0.132 | 4.311 +/- 0.815 | 2.442 +/- 0.788 |
| SGD with mom(0.9) | 5.206 +/- 0.935 | 4.974 +/- 1.657 | 3.395 +/- 0.357 | 2.471 +/- 0.69 | 3.496 +/- 0.636 | 3.499 +/- 0.903 | 2.927 +/- 0.569 | 2.453 +/- 0.057 | 4.931 +/- 0.974 | 4.506 +/- 1.304 |

Test accuracy for Sparse and Dense Networks using CIFAR-10

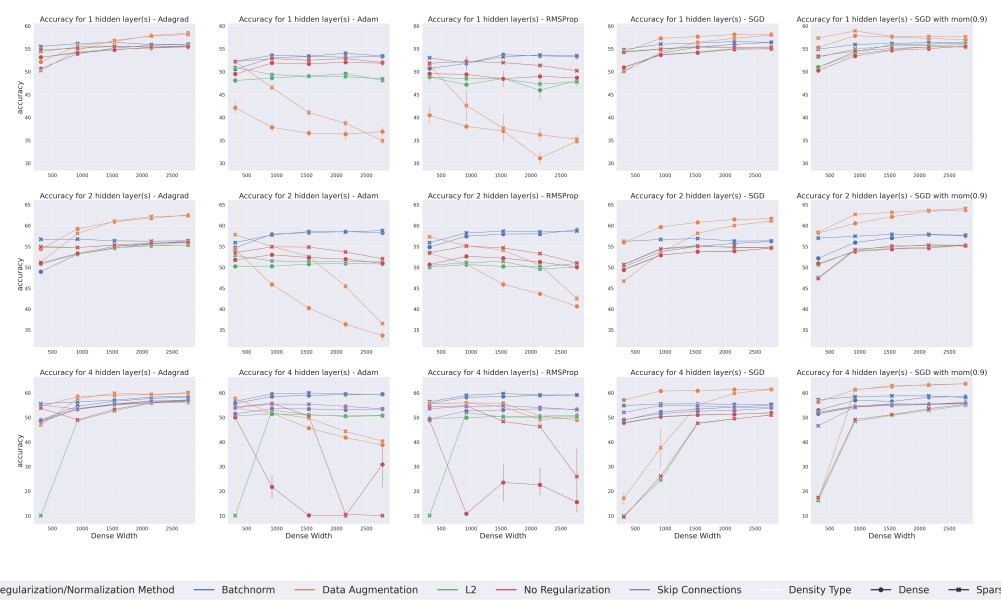

Figure 15: Test Accuracy for CIFAR-10 with 0.001 Learning Rate

Table 7: Test Accuracy summary for CIFAR-100 with low learning rate (0.001)

(a) One Hidden Layer

| | No Regularization | | Data Augmentation | | L2 | | Batchnorm | |
| | Dense | Sparse | Dense | Sparse | Dense | Sparse | Dense | Sparse |
|---|---|---|---|---|---|---|---|---|
| Adagrad | 26.76 +/- 0.886 | 27.39 +/- 0.644 | 26.958 +/- 2.287 | 26.22 +/- 2.659 | 27.023 +/- 1.035 | 27.588 +/- 0.874 | 21.491 +/- 1.413 | 22.974 +/- 0.299 |
| Adam | 23.016 +/- 1.688 | 24.185 +/- 0.657 | 13.401 +/- 1.361 | 16.3 +/- 4.214 | 21.771 +/- 1.059 | 22.191 +/- 0.493 | 24.255 +/- 2.204 | 25.111 +/- 0.906 |
| RMSProp | 22.115 +/- 1.05 | 23.764 +/- 0.737 | 12.805 +/- 1.663 | 15.415 +/- 3.953 | 20.953 +/- 1.012 | 22.063 +/- 0.747 | 24.009 +/- 2.011 | 24.715 +/- 1.092 |
| SGD | 26.911 +/- 1.177 | 27.043 +/- 1.958 | 26.525 +/- 1.234 | 24.081 +/- 3.158 | 27.051 +/- 1.155 | 27.086 +/- 2.041 | 21.014 +/- 2.475 | 23.002 +/- 0.659 |
| SGD with mom(0.9) | 25.155 +/- 2.71 | 26.243 +/- 1.264 | 29.007 +/- 1.125 | 29.577 +/- 1.318 | 25.663 +/- 2.55 | 26.632 +/- 1.299 | 21.649 +/- 2.645 | 23.565 +/- 0.631 |

(b) Two Hidden Layers

| | No Regularization | | Data Augmentation | | L2 | | Batchnorm | |
| | Dense | Sparse | Dense | Sparse | Dense | Sparse | Dense | Sparse |
|---|---|---|---|---|---|---|---|---|
| Adagrad | 28.21 +/- 1.497 | 28.748 +/- 1.701 | 28.773 +/- 2.741 | 27.423 +/- 4.207 | 28.495 +/- 1.572 | 29.063 +/- 1.805 | 24.932 +/- 3.263 | 27.318 +/- 1.216 |
| Adam | 20.423 +/- 1.323 | 23.28 +/- 2.231 | 12.156 +/- 4.33 | 17.349 +/- 6.332 | 20.977 +/- 0.507 | 22.596 +/- 1.217 | 28.15 +/- 1.968 | 28.558 +/- 1.696 |
| RMSProp | 18.585 +/- 2.791 | 21.013 +/- 4.295 | 12.551 +/- 4.858 | 16.499 +/- 5.901 | 21.065 +/- 0.566 | 21.967 +/- 1.24 | 28.151 +/- 2.113 | 28.132 +/- 1.841 |
| SGD | 27.125 +/- 1.31 | 24.099 +/- 7.827 | 27.951 +/- 1.625 | 22.25 +/- 6.112 | 27.271 +/- 1.262 | 24.039 +/- 7.938 | 23.861 +/- 4.083 | 26.622 +/- 1.036 |
| SGD with mom(0.9) | 26.972 +/- 2.252 | 26.289 +/- 3.761 | 31.985 +/- 2.717 | 31.633 +/- 2.95 | 27.328 +/- 2.375 | 26.621 +/- 3.86 | 25.421 +/- 3.022 | 27.196 +/- 1.268 |

(c) Four Hidden Layers

| | No Regularization | | Data Augmentation | | L2 | | Batchnorm | | Skip Connections | |
| | Dense | Sparse | Dense | Sparse | Dense | Sparse | Dense | Sparse | Dense | Sparse |
|---|---|---|---|---|---|---|---|---|---|---|
| Adagrad | 24.956 +/- 1.686 | 22.189 +/- 2.633 | 25.479 +/- 1.175 | 22.624 +/- 3.991 | 24.556 +/- 1.343 | 16.263 +/- 8.572 | 27.008 +/- 4.51 | 29.011 +/- 2.027 | 26.852 +/- 1.781 | 27.279 +/- 1.334 |
| Adam | 15.583 +/- 3.736 | 22.713 +/- 3.209 | 8.865 +/- 3.933 | 10.306 +/- 6.494 | 19.646 +/- 0.889 | 17.555 +/- 8.751 | 29.652 +/- 1.801 | 30.013 +/- 1.688 | 23.748 +/- 1.504 | 25.151 +/- 0.657 |
| RMSProp | 4.97 +/- 8.083 | 17.983 +/- 9.803 | 11.597 +/- 7.201 | 20.396 +/- 3.643 | 19.835 +/- 0.637 | 16.678 +/- 9.06 | 29.501 +/- 1.853 | 29.667 +/- 1.86 | 21.43 +/- 7.297 | 21.363 +/- 9.074 |
| SGD | 20.809 +/- 0.911 | 11.776 +/- 8.53 | 26.802 +/- 1.492 | 14.004 +/- 10.133 | 20.695 +/- 0.743 | 11.375 +/- 8.282 | 24.511 +/- 3.541 | 27.515 +/- 1.428 | 23.805 +/- 1.551 | 22.871 +/- 7.053 |
| SGD with mom(0.9) | 25.269 +/- 1.456 | 17.466 +/- 8.93 | 29.958 +/- 3.146 | 24.143 +/- 12.147 | 24.741 +/- 1.38 | 16.867 +/- 8.621 | 27.301 +/- 3.007 | 28.893 +/- 1.526 | 26.817 +/- 1.83 | 25.595 +/- 3.516 |

Table 8: Test Loss summary for CIFAR-100 with low learning rate (0.001)

(a) One Hidden Layer

| | No Regularization | | Data Augmentation | | L2 | | Batchnorm | |
| | Dense | Sparse | Dense | Sparse | Dense | Sparse | Dense | Sparse |
|---|---|---|---|---|---|---|---|---|
| Adagrad | 3.715 +/- 0.262 | 3.581 +/- 0.301 | 3.147 +/- 0.1 | 3.177 +/- 0.116 | 3.631 +/- 0.215 | 3.491 +/- 0.255 | 3.922 +/- 0.107 | 3.846 +/- 0.171 |
| Adam | 440.694 +/- 189.721 | 342.236 +/- 207.693 | 38.232 +/- 23.824 | 84.171 +/- 110.671 | 46.989 +/- 22.993 | 26.927 +/- 22.742 | 15.53 +/- 2.784 | 15.77 +/- 2.959 |
| RMSProp | 261.914 +/- 81.721 | 358.131 +/- 234.544 | 39.084 +/- 24.3 | 743.694 +/- 668.044 | 37.436 +/- 14.38 | 48.654 +/- 47.815 | 15.824 +/- 1.809 | 16.68 +/- 1.186 |
| SGD | 3.655 +/- 0.03 | 3.305 +/- 0.141 | 3.184 +/- 0.062 | 3.283 +/- 0.134 | 3.617 +/- 0.03 | 3.289 +/- 0.13 | 4.015 +/- 0.101 | 3.867 +/- 0.052 |
| SGD with mom(0.9) | 7.038 +/- 1.577 | 5.627 +/- 0.09 | 3.538 +/- 0.221 | 3.373 +/- 0.195 | 5.608 +/- 1.201 | 4.62 +/- 0.016 | 5.008 +/- 0.495 | 4.609 +/- 0.039 |

(b) Two Hidden Layers

| | No Regularization | | Data Augmentation | | L2 | | Batchnorm | |
| | Dense | Sparse | Dense | Sparse | Dense | Sparse | Dense | Sparse |
|---|---|---|---|---|---|---|---|---|
| Adagrad | 4.687 +/- 0.772 | 4.178 +/- 0.871 | 3.128 +/- 0.081 | 3.154 +/- 0.152 | 4.351 +/- 0.559 | 3.884 +/- 0.623 | 4.17 +/- 0.129 | 3.784 +/- 0.154 |
| Adam | 441.033 +/- 76.035 | 493.788 +/- 215.952 | 4.593 +/- 0.368 | 5.09 +/- 1.28 | 12.361 +/- 0.424 | 9.121 +/- 2.275 | 17.406 +/- 4.665 | 17.608 +/- 4.662 |
| RMSProp | 346.064 +/- 99.04 | 460.908 +/- 212.828 | 4.157 +/- 0.293 | 6.024 +/- 1.985 | 13.274 +/- 0.776 | 11.817 +/- 3.779 | 16.036 +/- 2.394 | 17.701 +/- 1.539 |
| SGD | 4.395 +/- 0.151 | 3.551 +/- 0.445 | 3.048 +/- 0.077 | 3.327 +/- 0.31 | 4.303 +/- 0.142 | 3.53 +/- 0.434 | 4.293 +/- 0.714 | 3.733 +/- 0.025 |
| SGD with mom(0.9) | 8.28 +/- 2.146 | 7.393 +/- 1.534 | 6.507 +/- 1.228 | 4.739 +/- 1.349 | 6.048 +/- 1.465 | 5.623 +/- 1.288 | 5.096 +/- 0.873 | 4.337 +/- 0.068 |

(c) Four Hidden Layers

| | No Regularization | | Data Augmentation | | L2 | | Batchnorm | | Skip Connections | |
| | Dense | Sparse | Dense | Sparse | Dense | Sparse | Dense | Sparse | Dense | Sparse |
|---|---|---|---|---|---|---|---|---|---|---|
| Adagrad | 9.714 +/- 3.113 | 10.018 +/- 5.165 | 4.584 +/- 1.222 | 4.181 +/- 1.036 | 9.112 +/- 2.96 | 10.83 +/- 4.376 | 4.275 +/- 0.491 | 3.784 +/- 0.042 | 7.227 +/- 2.173 | 6.592 +/- 2.377 |
| Adam | 209.854 +/- 400.362 | 10867.689 +/- 10783.68 | 5.044 +/- 0.288 | 8.551 +/- 3.795 | 32.786 +/- 12.029 | 15.217 +/- 12.419 | 19.798 +/- 6.003 | 20.605 +/- 6.057 | 779.235 +/- 319.831 | 715.685 +/- 395.333 |
| RMSProp | 222.053 +/- 450.311 | 3680.784 +/- 5265.27 | 14.482 +/- 16.75 | 5.609 +/- 0.51 | 22.649 +/- 5.115 | 15.09 +/- 9.165 | 20.313 +/- 5.053 | 21.787 +/- 3.96 | 649.551 +/- 366.815 | 443.976 +/- 348.581 |
| SGD | 10.394 +/- 1.219 | 5.936 +/- 2.56 | 3.082 +/- 0.064 | 3.786 +/- 0.619 | 10.222 +/- 1.053 | 5.961 +/- 2.521 | 4.286 +/- 0.652 | 3.704 +/- 0.128 | 8.212 +/- 0.61 | 4.608 +/- 1.706 |
| SGD with mom(0.9) | 13.425 +/- 2.123 | 12.899 +/- 4.883 | 7.215 +/- 0.572 | 5.239 +/- 1.298 | 9.114 +/- 1.778 | 9.882 +/- 3.843 | 4.724 +/- 0.861 | 4.135 +/- 0.203 | 10.16 +/- 1.981 | 8.783 +/- 0.955 |

Table 9: Test Accuracy summary for CIFAR-10 with high learning rate (0.1)

(a) Four Hidden Layers

| | No Regularization | | Data Augmentation | | L2 | | Batchnorm | | Skip Connections | |
| | Dense | Sparse | Dense | Sparse | Dense | Sparse | Dense | Sparse | Dense | Sparse |
|---|---|---|---|---|---|---|---|---|---|---|
| Adagrad | 19.469 +/- 10.563 | 31.377 +/- 15.801 | 22.62 +/- 16.083 | 37.885 +/- 17.664 | 19.739 +/- 12.699 | 40.133 +/- 14.52 | 56.194 +/- 1.319 | 57.536 +/- 0.712 | 15.615 +/- 9.853 | 30.756 +/- 17.113 |
| Adam | 10.0 +/- 0.0 | 10.018 +/- 0.005 | 10.0 +/- 0.0 | 10.0 +/- 0.0 | 9.998 +/- 0.012 | 9.98 +/- 0.036 | 53.191 +/- 2.687 | 54.964 +/- 1.88 | 9.999 +/- 0.003 | 10.001 +/- 0.005 |
| RMSProp | 9.999 +/- 0.0 | 10.001 +/- 0.01 | 10.0 +/- 0.0 | 10.001 +/- 0.003 | 10.0 +/- 0.0 | 10.191 +/- 0.578 | 53.328 +/- 1.022 | 53.62 +/- 1.056 | 10.0 +/- 0.004 | 10.001 +/- 0.0011 |
| SGD | 57.603 +/- 1.546 | 56.487 +/- 3.866 | 51.81 +/- 6.396 | 60.065 +/- 1.465 | 8.255 +/- 19.659 | 30.631 +/- 26.518 | 59.191 +/- 1.805 | 60.295 +/- 0.949 | 57.599 +/- 1.302 | 56.398 +/- 2.893 |
| SGD with mom(0.9) | 9.303 +/- 2.608 | 12.872 +/- 14.06 | 10.0 +/- 0.0 | 19.086 +/- 18.81 | 10.579 +/- 10.646 | 35.2 +/- 23.74 | 57.822 +/- 1.012 | 59.058 +/- 0.886 | 9.334 +/- 2.582 | 9.117 +/- 9.037 |

Table 10: Test Loss summary for CIFAR-10 with high learning rate (0.1)

(a) Four Hidden Layers

| | No Regularization | | Data Augmentation | | L2 | | Batchnorm | | Skip Connections | |
| | Dense | Sparse | Dense | Sparse | Dense | Sparse | Dense | Sparse | Dense | Sparse |
|---|---|---|---|---|---|---|---|---|---|---|
| Adagrad | 4.332 +/- 3.705 | 6.299 +/- 3.469 | 2.036 +/- 0.344 | 1.895 +/- 0.263 | 3.21 +/- 1.888 | 3.186 +/- 1.288 | 5.979 +/- 0.403 | 5.19 +/- 1.109 | 3.934 +/- 2.999 | 5.788 +/- 3.694 |
| Adam | 2.328 +/- 0.066 | 6.323 +/- 11.003 | 2.303 +/- 0.0 | 2.303 +/- 0.0 | 405749.405 +/- 1051430.43 | 343515.597 +/- 1330067.381 | 51.96 +/- 34.783 | 60.026 +/- 18.807 | 4.174 +/- 7.207 | 2.366 +/- 0.17 |
| RMSProp | 1339.999 +/- 5147.876 | 275.11 +/- 831.374 | 2.303 +/- 0.0 | 2.303 +/- 0.002 | 212.06 +/- 812.388 | 43851.25 +/- 122539.555 | 40.846 +/- 17.869 | 107.835 +/- 42.366 | 30.677 +/- 104.926 | 349.709 +/- 1315.569 |
| SGD | 6.609 +/- 1.155 | 7.01 +/- 2.525 | 4.964 +/- 1.373 | 3.744 +/- 1.348 | 0.339 +/- 0.897 | 1.659 +/- 1.237 | 3.758 +/- 0.389 | 3.254 +/- 0.341 | 4.634 +/- 0.791 | 4.674 +/- 1.132 |
| SGD with mom(0.9) | 2.138 +/- 0.615 | 3.166 +/- 4.273 | 2.303 +/- 0.0 | 2.129 +/- 0.359 | 1.797 +/- 0.975 | 2.175 +/- 0.895 | 6.208 +/- 0.244 | 5.396 +/- 0.905 | 2.149 +/- 0.595 | 4.388 +/- 8.928 |

Test accuracy for Sparse and Dense Networks using CIFAR-100

Figure 16: Test Accuracy for CIFAR-100 with 0.001 Learning Rate

Test accuracy for Sparse and Dense Networks using CIFAR-10

Figure 17: Test Accuracy for CIFAR-10 with 0.1 Learning Rate

Table 11: Test Accuracy summary for CIFAR-100 with high learning rate (0.1)

(a) Four Hidden Layers

| | No Regularization | | Data Augmentation | | L2 | | Batchnorm | | Skip Connections | |
|---|---|---|---|---|---|---|---|---|---|---|
| | Dense | Sparse | Dense | Sparse | Dense | Sparse | Dense | Sparse | Dense | Sparse |
| Adagrad | 4.832 +/- 3.339 | 11.731 +/- 7.078 | 10.304 +/- 7.42 | 14.827 +/- 8.334 | 5.383 +/- 5.543 | 14.899 +/- 9.16 | 28.804 +/- 1.905 | 29.826 +/- 1.857 | 5.885 +/- 3.829 | 11.649 +/- 6.443 |
| Adam | 1.0 +/- 0.0 | 1.0 +/- 0.0 | 1.0 +/- 0.0 | 1.0 +/- 0.0 | 1.0 +/- 0.0 | 0.999 +/- 0.007 | 22.995 +/- 1.308 | 23.792 +/- 2.673 | 1.0 +/- 0.0 | 1.001 +/- 0.003 |
| RMSProp | 1.0 +/- 0.0 | 1.001 +/- 0.004 | 1.0 +/- 0.0 | 1.0 +/- 0.0 | 1.0 +/- 0.0 | 1.0 +/- 0.008 | 24.424 +/- 1.281 | 25.13 +/- 1.514 | 0.999 +/- 0.003 | 0.999 +/- 0.005 |
| SGD | 25.719 +/- 11.926 | 23.833 +/- 11.53 | 11.46 +/- 5.91 | 22.79 +/- 10.489 | 28.546 +/- 8.056 | 24.077 +/- 11.917 | 30.83 +/- 2.898 | 31.749 +/- 1.631 | 26.638 +/- 9.891 | 28.838 +/- 3.675 |
| SGD with mom(0.9) | 1.0 +/- 0.0 | 5.53 +/- 6.788 | 1.0 +/- 0.0 | 8.005 +/- 9.069 | 4.471 +/- 5.342 | 12.283 +/- 10.29 | 31.041 +/- 2.264 | 31.778 +/- 1.709 | 1.003 +/- 0.009 | 1.689 +/- 2.477 |

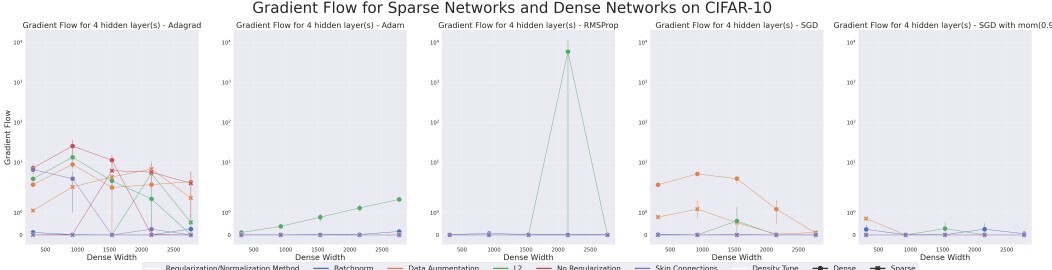

Figure 18: Gradient Flow for CIFAR-10 with 0.1 Learning Rate

Table 12: Test Loss summary for CIFAR-100 with low learning rate (0.1)

(a) Four Hidden Layers

| | No Regularization | | Data Augmentation | | L2 | | Batchnorm | | Skip Connections | |
| | Dense | Sparse | Dense | Sparse | Dense | Sparse | Dense | Sparse | Dense | Sparse |
|---|---|---|---|---|---|---|---|---|---|---|
| Adagrad | 7.667 +/- 4.947 | 14.836 +/- 8.468 | 4.13 +/- 0.358 | 5.685 +/- 2.939 | 6.024 +/- 3.425 | 6.571 +/- 2.298 | 9.474 +/- 0.926 | 8.717 +/- 1.854 | 9.247 +/- 5.927 | 15.916 +/- 9.915 |
| Adam | 4.664 +/- 0.201 | 8.052 +/- 8.361 | 4.605 +/- 0.0 | 4.605 +/- 0.0 | 27.215 +/- 87.569 | 64.341 +/- 137.011 | 170.973 +/- 84.218 | 367.302 +/- 141.566 | 4.631 +/- 0.081 | 4.752 +/- 0.508 |
| RMSProp | 11.792 +/- 23.242 | 57.154 +/- 202.471 | 4.605 +/- 0.0 | 4.606 +/- 0.003 | 4.605 +/- 0.0 | 5404.823 +/- 14437.166 | 146.552 +/- 73.377 | 748.996 +/- 361.425 | 5734.776 +/- 20650.808 | 5.919 +/- 3.307 |
| SGD | 10.48 +/- 2.908 | 11.058 +/- 3.433 | 9.797 +/- 5.488 | 11.97 +/- 5.69 | 5.7 +/- 1.647 | 5.322 +/- 0.59 | 5.587 +/- 0.934 | 5.004 +/- 0.265 | 11.017 +/- 7.566 | 8.611 +/- 1.653 |
| SGD with mom(0.9) | 4.605 +/- 0.0 | 12.49 +/- 11.831 | 4.605 +/- 0.0 | 4.367 +/- 0.342 | 3.903 +/- 1.591 | 4.965 +/- 0.518 | 10.612 +/- 1.41 | 9.014 +/- 2.289 | 4.607 +/- 0.005 | 16.577 +/- 29.665 |

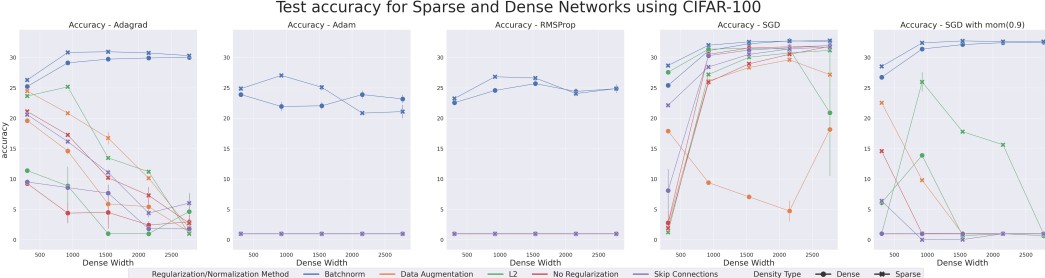

Figure 19: Test Accuracy for CIFAR-100 with 0.1 Learning Rate

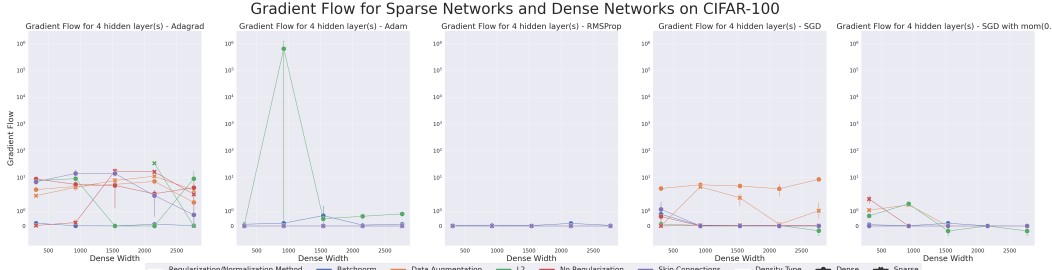

Figure 20: Gradient Flow for CIFAR-100 with 0.1 Learning Rate

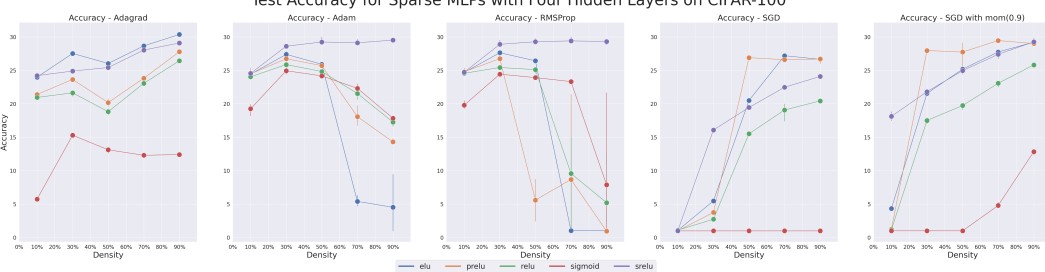

Figure 21: Test Accuracy for Different Activation Functions

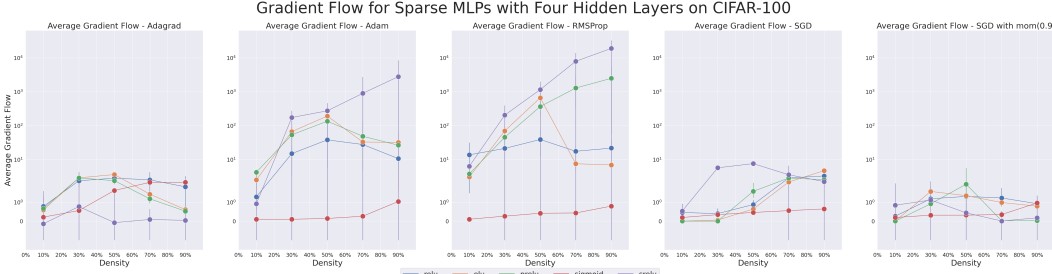

Figure 22: Gradient Flow for Different Activation Functions

