# OpenReview forum: "Keep the Gradients Flowing: Using Gradient Flow to study Sparse Network Optimization"
_ICLR.cc/2021/Conference — Reject_

### Official Review · AnonReviewer4 · 2020-10-25
**Meaningful investigation on the impact of different factors in sparse training**

**Rating:** 5
**Confidence:** 3

**Review:**

Recently, initialization has been found critical to the model accuracy attained by sparse training [1]. In this paper, the authors studies the impact of factors other than initialization on the model accuracy attained by sparse training relative to dense training (under the same model parameter count). At the core of this paper, the authors argue that the effective gradient flow (grad norm from only activate model weight dimensions) is an effective indicator on the model accuracy attained by sparse training. Firstly, the authors show that the effective gradient flow attains higher correlation with model accuracy than the norm of full gradient (including gradients on sparsified weight dimensions) in sparse training. Secondly, the paper empirically demonstrate that in sparse training, 1) weight decay and data augmentation can hurt model accuracy, 2) batch normalization plays significantly role for model accuracy in sparse training and 3) non-saturating activations boost the magnitude of effective gradient flow and consequently improve model accuracy.

I think the direction this paper worked on can have meaningful influence on designing model structure and general training algorithms for sparse training. Specifically, the effective gradient flow can potentially be an objective to improve for innovate methods in sparse training. Additionally, the investigation presented the impact of different model and training factors on model accuracy that has not been systematically studied before; this could bring up discussions for more broader techniques in sparse training.


Currently I give marginal reject because I have the following comments on technical aspect and clarity regarding meeting the acceptance criteria. I am happy to raise the rating if the authors can resolve them during the rebuttal phase.

1. In Table 1. the authors show that effective gradient flow achieves higher correlation with model accuracy than full grad norm. However the difference in the correlation attained by effective gradient flow and full grad norm is not significant. I was wondering which correlation is used here? Is it the Pearson Coefficient for linear correlation or rank correlation? If the rank correlation is not used here, I would suggest the authors to present results on rank correlation. Given that the effective gradient flow and model accuracy might not follow a linear relationship, the rank correlation can help better evaluate whether better effective gradient flow tends to imply better model accuracy in sparse training.

2.  In section 4.1, how are the training hyperparameter tuned for different methods such as using and without using weight decay? Some quick elaboration on the fairness for tuning each setting will be useful to help evaluation the conclusion drawn in section 4.1. Also is the model trained for enough number of optimizer steps to attain almost-saturating accuracy for each specific setting?

3. In section 4.1, it is a bit surprising to me that data augmentation can hurt generalization performance in sparse training. Are there any intuitive reasons for this observation?


Besides the above major comments, here are the minor comments for further improving the paper:

1. The definition and use of symbol a_s is confusing. In Equ. (2), a_s is a deterministic scalar value while in Equ. (3) a_s becomes a random variable.

2. It would be helpful to clarify why the authors choose to use 1000 epochs (or is it 1000 optimizer steps?) uniformly in the experiments; this can help readers to better evaluate the experiment protocols.

3. Symbol m in Equ. (6) should be bold to be consistent with vector notations on gradient.

4. Larger font in figures will be useful for smooth reading.

5. I totally understand that in section 4.1 that MLP for CIFAR100 is a micro experiment for study and < 40% accuracy can be acceptable. But giving dense training accuracy numbers/curves under the same model parameter count can be helpful better evaluating the sparse training accuracy relative to dense training.


Reference
[1] The Lottery Ticket Hypothesis: Finding Sparse, Trainable Neural Networks. Jonathan Frankle, Michael Carbin. ICLR 2018.

---

> ### Author Response · Authors · 2020-11-19
> **Thanks for constructive review!**
>
> We thank R4 for their review and appreciate the feedback as it helped us clarify and improve the quality of our paper. We hope with the improvements, we can address your concerns. If not, we look forward to a productive discussion.
>
> ----->In Table 1. the authors show that effective gradient flow achieves higher correlation with model accuracy than full grad norm. However the difference in the correlation attained by effective gradient flow and full grad norm is not significant. I was wondering which correlation is used here? Is it the Pearson Coefficient for linear correlation or rank correlation? If the rank correlation is not used here, I would suggest the authors to present results on rank correlation. Given that the effective gradient flow and model accuracy might not follow a linear relationship, the rank correlation can help better evaluate whether better effective gradient flow tends to imply better model accuracy in sparse training.
>
> We thank the reviewer for the suggestion. We used Kendall Rank correlation in our comparison. We have made this apparent in the paper, in the description under table 1.
>
> ----->In section 4.1, how are the training hyperparameter tuned for different methods such as using and without using weight decay? Some quick elaboration on the fairness for tuning each setting will be useful to help evaluation the conclusion drawn in section 4.1. Also is the model trained for enough number of optimizer steps to attain almost-saturating accuracy for each specific setting?
>
> We appreciate this opportunity to clarify on our hyperparameter choices.
>
> For weight decay, we use 0.0001 as the decay coefficient. We did a crude search of common weight decay parameters and found this parameter to work best.
> For data augmentation, we use the standard data augmentation procedure used for each dataset (random flips and crops).
> For batchnorm, we use the version that considers gamma and beta as constants (affine=False in pytorch), as opposed to learnable parameters to ensure no advantage to the sparse variants.
> For skip connections, we add skip connections every two layers as was done in the original paper [1].
> We trained the networks for 1000 epochs (approximately 450 000 steps), which was enough to ensure all network’s performance converged.
>
> ----->In section 4.1, it is a bit surprising to me that data augmentation can hurt generalization performance in sparse training. Are there any intuitive reasons for this observation?
>
> We thank R4 for drawing our attention to this. We believe this is because data augmentation (DA) methods have been designed for convolutional networks (CNNs) and not for multilayer perceptrons (MLPs). For MLPs, DA might not have the intended effect since MLPs work with a flattened image, across all the channels. This means random crops and flips might not have the same effect as they do for CNNs, which leverage pooling and 3d conv layers.
>
> In subsequent revisions, we will include a discussion of this in the manuscript.
>
> ----->The definition and use of symbol a_s is confusing. In Equ. (2), a_s is a deterministic scalar value while in Equ. (3) a_s becomes a random variable.
>
> We thank R4 for pointing this out.  We have refined the equations in section 2.1 and added more details in the appendix, section A1.
>
> ----->It would be helpful to clarify why the authors choose to use 1000 epochs (or is it 1000 optimizer steps?) uniformly in the experiments; this can help readers to better evaluate the experiment protocols.
>
> Experimentally, we found that 1000 epochs (approximately 450 000 steps) was more than enough training time to ensure all networks converged. Having fewer epochs (not ensuring convergence) and using methods like early stopping, we believe would have detracted from a fair comparison.
>
> ----->Symbol m in Equ. (6) should be bold to be consistent with vector notations on gradient.
>
> We have corrected this.
>
> ----->Larger font in figures will be useful for smooth reading.
>
> In our subsequent revision, we will ensure text in images are more legible.
>
> [1] He, K., Zhang, X., Ren, S. and Sun, J., 2016. Deep residual learning for image recognition. In Proceedings of the IEEE conference on computer vision and pattern recognition (pp. 770-778).

---

> > ### Author Response · Authors · 2020-11-25
> > **Thanks for constructive review! - Part 2**
> >
> > We thank R4 for this opportunity to address more of your previous concerns.
> >
> > To better answer Q3, we have included a discussion in section 4.1, showing that DA only hurts generalization performance when used without batch normalization.

---

### Official Review · AnonReviewer3 · 2020-10-28
**claims not well supported**

**Rating:** 3
**Confidence:** 3

**Review:**

This paper proposes effective gradient flow (EGF), which is a layer-wise normalized gradient flow. Compared to (unnormalized) gradient flow, the paper shows that the proposed EGF is (slightly) better correlated with metrics like test loss and test accuracy (see Table 1). Given that this claim is supported with experimental results, the paper would become much stronger if a larger number and a more diverse set of data-sets were used (in addition to CIFAR-10 and CIFAR-100, which are two very similar image-data-sets) as to show that the claim holds more generally. Apart from that, given that the correlations are (only) about 0.4 in Table 1, it seems that only some aspects are explained by EGF.

This said, if we continue with the insight that EGF is helpful, how would EGF help to design better sparse models and optimization methods? Why is it easier to compute EGF instead of directly the relevant metrics (like test loss etc.) to determine what training methods and model architectures work better?  In fact, Figure 2, top row, exactly does this: test-accuracy is directly used to determine which training methods work best for sparse models. What is the additional insight/benefit of using EGF (like in the bottom row of Figure 2) instead of test-accuracy ?

The paper also proposes the SC-SDC framework, see also results in Table 3. The key idea is to compare  sparse and dense networks that have the same number of (non-zero) weights. While this is a good start, I am not sure that this is really a fair comparison, though. I would expect the sparse network to have a larger capacity than a dense network with the same number of parameters. While there are many ways to see this, the simplest  might be that  a weight in a sparse model requires two parameters, its value and its index if we wanted to encode the model. Another way to see it might be that a dense model can be pruned to become sparse(with fewer weights), but without losing much prediction accuracy. As a consequence of potentially comparing sparse and dense networks of different capacities, the results in Table 3 might be biased in favor of sparse models. From this perspective, it is remarkable that sparse models do not clearly outperform dense models in Table 3, which might indicate that the training of sparse models with the analyzed approaches still suffers from the problems outlined earlier in the paper.

Equipped with the proposed methods EGF and SC-SDF, the paper then analyzes several (standard) approaches like batch normalization etc as to determine which of these approaches are useful for training sparse models.  Again, given that only two very similar data-sets were used (CIFAR-10 and 100), it is unclear if the found results would generalize to other data sets. Moreover, it is not clear why EGF and SC-SDF are needed to determine which training-methods work well for sparse models. In fact, Figure 2 (top row) directly shows the test-accuracy for the various approaches--i.e., without using the proposed EGF and SC-SDF. EGF is shown in the bottom row in Figure 2, as to illustrate that EGF has a similar behavior as test accuracy in the first row in Figure 2. What is the additional insight obtained from EGF compared to using only test-accuracy as to determine that  batch-normalization works well ?

Some minor points:

How are “test loss“ and “test accuracy” defined in this paper, and what is the difference? I did not immediately find the answer in the referenced paper (Jiang 2019).

Are really all 6 digits statistically significant in Table 1 ?

Figures 7 and 9 in the appendix are missing the actual image.

The paper has several typos, like “and.”, or the first sentence in Appendix A

+++ updates after authors' feedback +++

I  thank the reviewers for their detailed response. I still feel that more work (experimental and theoretical, as outlined in my review) is needed.

I also would like to make sure that my point is not misunderstood when I said that the sparse model requires twice as many parameters to be stored (the value and the index),  compared to the dense model. Using compression algorithms, the number of bits to store the model can obviously be reduced (below the factor of two). Anyways, the deeper point that I wanted to make  was the connection between minimum description length  (number of bits to store the model) in information theory and the model complexity /capacity in statistics/machine learning: see BIC in https://en.wikipedia.org/wiki/Minimum_description_length
Hence, the bits to store a model are directly related to the model-complexity/capacity in machine-learning/statistics.

---

> ### Author Response · Authors · 2020-11-19
> **Thanks for your review. Reply 1 (Part I/II)**
>
> We thank R3 for the suggestions and insightful review. We will address your comments in two parts. Firstly, we will address the questions about the details of our methodology. Secondly, in a few days, we will address broader questions about the benefits of EGF and SC-SDC.
>
> ----->Compared to (unnormalized) gradient flow, the paper shows that the proposed EGF is (slightly) better correlated with metrics like test loss and test accuracy (see Table 1). Given that this claim is supported with experimental results, the paper would become much stronger if a larger number and a more diverse set of data-sets were used (in addition to CIFAR-10 and CIFAR-100, which are two very similar image-data-sets) as to show that the claim holds more generally.
>
> We thank the reviewer for this suggestion. We do agree that extending the experiments to more datasets would be beneficial to supporting the EGF formulation. Running these thorough experiments on large datasets like ImageNet is not feasible for us, but we have extended these experiments to Fashion MNIST, a dataset that is considerably different to the CIFAR datasets. We will include these results in Appendix B.
>
> ----->Apart from that, given that the correlations are (only) about 0.4 in Table 1, it seems that only some aspects are explained by EGF.
>
> Although it is true that EGF cannot explain all parts of network performance, our goal is to show that it is a more feasible measure to study optimization as compared to previous measures. Furthermore, as far as we are aware, there is no single measure for deep neural networks that has had a direct correlation with performance.
>
> ----->The paper also proposes the SC-SDC framework, see also results in Table 3. The key idea is to compare sparse and dense networks that have the same number of (non-zero) weights. While this is a good start, I am not sure that this is really a fair comparison, though. I would expect the sparse network to have a larger capacity than a dense network with the same number of parameters. While there are many ways to see this, the simplest might be that a weight in a sparse model requires two parameters, its value and its index if we wanted to encode the model. Another way to see it might be that a dense model can be pruned to become sparse(with fewer weights), but without losing much prediction accuracy.
>
> We thank R3 for providing feedback on the SC-SDC framework. We would like to clarify that when we say sparse and dense networks have the same capacity, our notion of “same capacity” refers to the same number of weights/connections. Other notions of “same capacity” can also be considered and would be interesting in future work.
>
> It appears the reviewer’s first example is relating to the storage of sparse matrices. If sparse matrices are stored as a coordinate list, it is correct that you would need to store the index and value. However, this doesn’t relate to what we are referring to as capacity ( representational power) of the network. Irrespective of the storage, the sparse network has the same number of learnable, model parameters as the dense network. We have included a short clarification of this after point 4, in section 2.1. We have also included a brief discussion of this in Appendix A1.
>
> The second example refers to a pruned setting, where unimportant weights are removed, while maintaining or preserving accuracy. This is usually done by a process of training a dense model first and then using a pruning criteria to remove weights that are perceived to have the least impact on performance. In our case we are comparing random, fixed sparse networks to their dense counterparts. We have made this clearer in the introduction (paragraph 4).
>
>
> ----->As a consequence of potentially comparing sparse and dense networks of different capacities, the results in Table 3 might be biased in favor of sparse models. From this perspective, it is remarkable that sparse models do not clearly outperform dense models in Table 3, which might indicate that the training of sparse models with the analyzed approaches still suffers from the problems outlined earlier in the paper.
>
> Most neural network optimization and training routines have been catered and designed for dense networks. Furthermore, we are using static, random, sparse networks. We believe both these factors lead to sparse models not consistently outperforming dense ones.

---

> > ### Author Response · Authors · 2020-11-19
> > **Thanks for your review. Reply 1 (Part II/II)**
> >
> > ----->How are “test loss“ and “test accuracy” defined in this paper, and what is the difference? I did not immediately find the answer in the referenced paper (Jiang 2019).
> >
> > We use “test loss” and “test accuracy” to refer to the performance of the model on the test set. Test loss refers to the cross entropy loss, while test accuracy refers to what percentage of examples, the model correctly classifies in the test set.
> >
> > ----->Are really all 6 digits statistically significant in Table 1 ?
> >
> > We apologize for this. In our subsequent update, we will correct this.
> >
> > ----->The paper has several typos, like “and.”, or the first sentence in Appendix A
> >
> > We thank R3 for pointing our attention to these errors. We have now revised the paper and fixed these (and other) typos and added the missing figures to the appendix. We will also improve the clarity and make more updates to the Appendix in the next few days.

---

> > > ### Author Response · Authors · 2020-11-25
> > > **Thanks for your review! Reply 2 (Part I/I)**
> > >
> > > We thank R3 for this opportunity to address more of your previous concerns.
> > >
> > > -----> How would EGF help to design better sparse models and optimization methods? Why is it easier to compute EGF instead of directly the relevant metrics (like test loss etc.) to determine what training methods and model architectures work better? In fact, Figure 2, top row, exactly does this: test-accuracy is directly used to determine which training methods work best for sparse models. What is the additional insight/benefit of using EGF (like in the bottom row of Figure 2) instead of test-accuracy ?
> > >
> > > ----->Moreover, it is not clear why EGF and SC-SDF are needed to determine which training-methods work well for sparse models. In fact, Figure 2 (top row) directly shows the test-accuracy for the various approaches--i.e., without using the proposed EGF and SC-SDF. EGF is shown in the bottom row in Figure 2, as to illustrate that EGF has a similar behavior as test accuracy in the first row in Figure 2. What is the additional insight obtained from EGF compared to using only test-accuracy as to determine that batch-normalization works well ?
> > >
> > > Gradient flow (gradient norms) have been used to study network behaviour and as gradient preserving pruning criteria [1,2,3]. Although, top-line metrics such as accuracy can tell us which training methods achieve better performance, they don’t provide possible information as to why certain methods work/fail. We introduced EGF to provide a measure that could possibly explain why certain interventions hurt or improve sparse network performance. This was the case in the analysis of weight decay and data augmentation in section 4.1.
> > >
> > > We have also extended our experiments and analysis to show the benefits of EGF, namely:
> > > - For adaptive optimization methods that use an exponentially decaying average of past squared gradients, such as Adam and RMSProp, a high EGF correlates to poor performance. We can see this with weight decay (with and without batch normalization) and data augmentation ( without batch normalization) as discussed in section 4.1.
> > > - We have included a section of EGF’s favourable properties on page 4. Notably, it can extend current gradient flow analysis methods.
> > >
> > > We introduced SC-SDC as a small scale framework to allow us to disentangle which of the sparse network behaviour is a direct result of a network being sparse and to identify what optimization and regularization components need to be adapted for sparse networks. We also showed the results of this framework extend to more conventional architectures such as WideResNet-50.
> > >
> > >
> > > References:
> > > [1] Evci, U., Ioannou, Y.A., Keskin, C. and Dauphin, Y., 2020. Gradient Flow in Sparse Neural Networks and How Lottery Tickets Win. arXiv preprint arXiv:2010.03533.
> > > [2] Eldan, R. and Shamir, O., 2016, June. The power of depth for feedforward neural networks. In Conference on learning theory (pp. 907-940).
> > > [3] Hornik, K., Stinchcombe, M. and White, H., 1989. Multilayer feedforward networks are universal approximators. Neural networks, 2(5), pp.359-366.
> > > [4] Funahashi, K.I., 1989. On the approximate realization of continuous mappings by neural networks. Neural networks, 2(3), pp.183-192.

---

### Official Review · AnonReviewer2 · 2020-10-28
**Sparse Networks and Gradient Flow**

**Rating:** 5
**Confidence:** 4

**Review:**

Summary:

This work performs a systematic study to understand how various design choices affect the performance and the training behavior of sparse neural networks. In order to do this, it proposes a simple framework to compare sparse and dense networks (SC-SDC) and a simple common-sense metric to measure the gradient flow in sparse neural networks (Effective Gradient Flow). It is the first time when I see such study, and I believe that it can contribute to the way how we perceive sparse neural networks as it brings a new perspective.

Strong Points:

S1) The paper can guide the researcher or the practitioner to make good design choices with respect to sparse networks in a fast manner.

S2) The well-designed experiments show the counterintuitive fact that even static random generated sparse network can outperform dense networks at the same capacity (number of parameters) if proper settings which improve the gradient flow are made, e.g. PReLU or SReLU as activation function, batch normalization.

S3) The results suggest that sparse networks do not necessarily falls under the same behavior of their dense counterparts and highlight the importance of having specialized algorithms and methods for them.

S4) In general, the paper is clear and well written, with some exceptions which likely show that it has been developed until the very last moment before submission.


Weak Points:

W1) The paper address just static sparse networks. As static networks still have a limited capacity, I believe that it would be much more informative if (ideally) the same sparse random generated networks would be studied in the same conditions using also dynamic sparse training.

W2) The claim that comparing sparse networks with dense networks having the same number of parameters (thus fewer neurons) is fair does not sound completely fair to me. I agree that it is a tricky situation overall, but I would consider also the fact that more neurons mean more representational power and, thus, sparse networks have an "unfair" advantage from this perspective.

W3) The general appearance of the paper suffers from, probably, the last-minute submission.


While the paper has clearly added value, overall, I found it a bit immature for publication. During the rebuttal phase, I would recommend to the authors to address the above weak points and the following (minor) comments:

C1) Make it clear from the begin (not just in the appendix) that you are discussing about static sparse networks to avoid confusion. Auxiliary, perhaps you can add some details about the sparse networks initialization in Table 2.

C2) It is very interesting in Table 4, first row, that for this particular case the networks fail to converge in the two “extreme” cases (Dense and 20% density), while the middle density levels work fine. Do you have any idea why this happens? Same reasons for the sparse and the dense networks? Perhaps, this particular case, can help in developing more robust training methods for neural networks in general?

C3) The clarity of the Dense Width paragraph from page 5 can be improved

C4) Perhaps you can add also the dense network performance in Fig. 2?

C5) Try to perform a thorough proof-read of the paper. There are many typos (e.g. page 3, “layer in l”), and other sloppy written aspects (e.g. detail MAW, tables 11 and 12 have the same learning rate value, figures are missing from the appendix, mention somewhere in tables 9-12 the activation function, etc.)

---

> ### Author Response · Authors · 2020-11-19
> **Thank you for your informative review!**
>
> We thank R2 for their informative and constructive review. We appreciate this opportunity to reply accordingly. We hope with the improvements, we can address your concerns. If not, we look forward to a productive discussion.
>
> ----->W1
>
> Although we agree with the reviewer that studying dynamic sparsity using measures such as EGF and SC-SDC will be interesting, we believe as a first step it is critical to focus on static sparsity training dynamics. Once some of the principles of static, random sparsity have been explored, we believe this can be extended to studying dynamic sparsity.
>
> Furthermore, there is existing work that explores gradient flow (standard gradient norm, not EGF) in a dynamic sparse setting (in the context of pruning) [1].
>
>
> ----->W2
>
> We thank the reviewer for this very valuable point. It is true that even with the same number of connections, the sparse network will (sparsely) connect to more neurons than its dense counterpart. It is possible that the increased number of activations being used can lead to sparse networks having higher representational power, however most work on expressivity of neural networks looks at this from a depth perspective and proves certain depths of networks are universal approximators [2,3,4].
>
> To this end, we ensure these networks have the same depth. Our notion of “same capacity” relates to the same number of active weights, but we believe going forward it would be an interesting direction to explore the notion of similar capacity by ensuring a similar number of neurons.
>
> We have also included in the last paragraph of section 2.1 and in Appendix A1,  that other notions of “same capacity” can exist and that although we maintain equal parameter count, we do not maintain equal number of neurons.
>
> ----->W3
>
> We thank R2 for pointing this out and we continue to improve the general appearance of the paper.
>
> ----->C1
>
> We have now included this fact in the introduction (paragraph 4). Furthermore, we have also included more details about our sparse network initialization in the Appendix A.1.
>
> ----->C2
>
> This occurs due to vanishing or exploding gradients. We will include, in the coming days, more details on this phenomenon.
>
> ----->C3
>
> We have improved the clarity of the Dense Width paragraph in section 3, by adding an example. Furthermore, we have added more Dense Width details in Appendix A1 and we have also included Figure 4 in Appendix A1, to build more intuition on Dense Width.
>
> ----->C4
> We will do this in subsequent revisions, in the next few days.
>
> ----->C5
>
> We thank R2 for pointing our attention to these errors. We have now revised the paper and fixed these (and other) typos and added the missing figures to the appendix. We will also improve the clarity and make more updates to the Appendix in the next few days.
>
>
> References:
> [1] Evci, U., Ioannou, Y.A., Keskin, C. and Dauphin, Y., 2020. Gradient Flow in Sparse Neural Networks and How Lottery Tickets Win. arXiv preprint arXiv:2010.03533.
> [2] Eldan, R. and Shamir, O., 2016, June. The power of depth for feedforward neural networks. In Conference on learning theory (pp. 907-940).
> [3] Hornik, K., Stinchcombe, M. and White, H., 1989. Multilayer feedforward networks are universal approximators. Neural networks, 2(5), pp.359-366.
> [4] Funahashi, K.I., 1989. On the approximate realization of continuous mappings by neural networks. Neural networks, 2(3), pp.183-192.

---

### Official Review · AnonReviewer1 · 2020-10-29
**Interesting direction, though the particular approach may not be the most impactful**

**Rating:** 5
**Confidence:** 4

**Review:**

Summary: This paper looks at how choices in optimization affect how well you can train sparse networks. The authors come up with a new measure of effective gradient flow, which is important for good performance. They also compare sparse vs. dense networks across various optimizers, hyperparameters and activation functions, and find that batch norm and certain activation functions are beneficial for sparse networks

Pros
* It is important to understand what’s better for training of sparse networks. There are previous studies of the effect of overparameterization, but much less about sparse networks vs. dense networks at the same size.
* The authors did a multitude of experiments thoroughly to compare different training configurations

Cons
* While it is interesting to compare sparse network to their same capacity dense networks, this comparison is less practical because small dense networks are rarely used in practice.
* I’m unsure about the novelty or rationale of the EGF definition: shouldn’t masked weights already have a gradient of 0? What is the rationale for not weighing each layer’s gradient equally, even if some have more weights than others?
* How are the sparse connections determined? If it is random, then the findings are less applicable because people would use sparse networks found by a good pruning method rather than a random sparse network in practice.
* While table 1 shows that EGF is better than simple gradient norms, gradient flow measures in general do not seem that important to the rest of the paper: why do we need gradient flow when we can just look at accuracy? Is EGF used for anything besides as an observation? Also, it is unclear in figure 2 if high EGF can explain good performance or if they are just somewhat correlated.

Other questions:
* How do you specify the number of active connections if $a_D$ of a fully connected layer is determined by number of inputs * number of outputs? Do you adjust the number of inputs or outputs, or both? How does the procedure work with conv layers?
* Table 1: when do you take EGF measures - throughout training or at the beginning or end? Also, for the experiments in figure 2, how does EGF change over time and is it averaging EGF over training the best way to use this measure?
* Why use a Wide ResNet rather than a regular ResNet?
* Doesn’t data augmentation hurt dense networks more for adam?
* What sparsity level does table 3 use?

Additional suggestions
* Perhaps it would also be useful to see the effect of adjusting more common/simple hyperparameters such as learning rate, batch size, and momentum.
* I would like to see accuracies of the experiments in table 3: even though it is scientifically interesting to see where sparse networks do better relative to dense networks, ultimately the goal is to get the best performing sparse network in terms of absolute accuracy. Figure 2 shows that, but not for all the combinations.
* Minor: table 4 might be easier to visualize as a chart, similar to figure 2.
* Minor: “batch normalizationalization” typo on page 5, and figures 7 and 9 do not show up

Overall: This paper examines many experiments to improve the optimization of sparse networks, but I am not convinced that the approaches are the most effective (how important is EGF and comparing sparse to same size dense). Thus, I recommend rejection but am open to changing my stance based on the authors’ response.

---

> ### Author Response · Authors · 2020-11-19
> **Thanks for the thorough review! -  Reply 1 (Part I/II) -**
>
> We thank R1 for their thorough and detailed review, and appreciate this chance to address some of the outstanding questions. In our first reply, we will address the questions about the specifics of EGF and our experimental process. Subsequently, in a follow up reply, we will address some of the more detailed questions relating to the importance and practicality of EGF and SC-SDC.
>
> ----->“I’m unsure about the novelty or rationale of the EGF definition: shouldn’t masked weights already have a gradient of 0? What is the rationale for not weighing each layer’s gradient equally, even if some have more weights than others?”
>
> Masked weights do not necessarily have zero gradient since the partial derivative of the weight wrt the loss, is influenced by other weights and activations. Thereby a weight can be zero, but its gradient can be nonzero. This is the reason why weights have to be masked at each forward step and cannot simply be masked once. We thank the reviewer for drawing attention to this point and we have included clarification in the benefits of EGF section (section 2.2), point titled “Only gradients of active weights are used”.
>
> Secondly, we do weigh each layer's gradient equally. Based upon R1’s comments,  we also have updated the manuscript to make this more clear, in the benefits of EGF section (section 2.2), point titled “Gradient flow is evenly distributed across layers”.
>
> Relating to novelty of this measure, current approaches measuring gradient flow in sparse networks (in the context of pruning), use the full gradient norm or approximates of it [1,2,3], which would include the gradients of masked weights. We believe our measure could be used to extend on these existing works.
>
> ----->“How do you specify the number of active connections if aD of a fully connected layer is determined by number of inputs * number of outputs? Do you adjust the number of inputs or outputs, or both? How does the procedure work with conv layers?”
>
> We have a max width for our sparse networks and apply a mask to ensure it has the same number of connections as the dense counterpart. We don’t change the number of inputs or output, we alter the hidden layer size. Due to the constraints on manuscript length, we included details of this methodology in Appendix A.1. Furthermore, we have included Figure 3, in appendix A1 for a visual example of this process.
>
> The current SC-SDC framework is catered for MLPs. To adapt it to work for conv layers, we would follow a similar process, but we need to ensure a matched parameter count in the 3d dimension (for multiple channels). This will be addressed in extensions of this methodology. Note the experiments for WideResnet are sparsely masked, but we do not restrict ourselves to matching a parameter count of a dense network, but rather to a sparsity/density level.
>
> ----->Table 1: when do you take EGF measures - throughout training or at the beginning or end? Also, for the experiments in figure 2, how does EGF change over time and is it averaging EGF over training the best way to use this measure?
>
> We measure the gradient flow at 10 points throughout training, specifically at the end of epochs 0, 99. 199, 299, 399, 499, 599, 699, 799, 899 and 999. We have added these experimental details to the manuscript for clarity (second last paragraph, section 2.2).
>
> The change in EGF over time varies vastly depending on the optimizer. We will include some charts showing this in Appendix B. Relating to averaging EGF, we considered other formulations such as minimum, maximum and variance of EGF, but we found the average being the most informative formulation.
>
> ----->Why use a Wide ResNet rather than a regular ResNet?
>
> We used Wide ResNet since it is a larger, more complicated model than the standard ResNet-50. We believe this will be a more realistic test of sparse optimization in convolutional networks.
>
> ----->What sparsity level does table 3 use?
>
> These results are across all sparsity levels. Since the Wilcoxon signed rank test is paired, we simply ensure that the correct sparse and dense results are paired together in the statistical test. For clarity, we have included this in the description under table 3.
>
> ----->Perhaps it would also be useful to see the effect of adjusting more common/simple hyperparameters such as learning rate, batch size, and momentum.
>
> We do agree this would be interesting, but due to computational limitations, it is not currently feasible to explore all these variants. We will include results where we have tested different learning rates (0.1 and 0.001) in section 4 (in subsequent revisions).
>
> For momentum, we use the common value of 0.9 and we use 128 as a batch size as is common for CIFAR datasets. In future, we believe it would be interesting to explore other hyperparameters/variants.

---

> > ### Author Response · Authors · 2020-11-19
> > **Thanks for the thorough review! - Reply 1 (Part II/II)**
> >
> > ----->I would like to see accuracies of the experiments in table 3: even though it is scientifically interesting to see where sparse networks do better relative to dense networks, ultimately the goal is to get the best performing sparse network in terms of absolute accuracy. Figure 2 shows that, but not for all the combinations.
> >
> > The specific accuracy results are presented in Appendix C, table 7c. We have noted under the table that the full set of results can be found in Appendix C. In our subsequent revision, we will include a direct reference to the exact, updated table.
> >
> > ----->Minor: table 4 might be easier to visualize as a chart, similar to figure 2.
> >
> > We thank R1 for the suggestion. We will be making the change accordly in our subsequent revision.
> >
> >
> > Reference
> > [1] Evci, U., Ioannou, Y.A., Keskin, C. and Dauphin, Y., 2020. Gradient Flow in Sparse Neural Networks and How Lottery Tickets Win. arXiv preprint arXiv:2010.03533.
> >
> > [2] Wang, C., Zhang, G. and Grosse, R., 2020. Picking winning tickets before training by preserving gradient flow. arXiv preprint arXiv:2002.07376.
> >
> > [3] Singh Lubana, E. and Dick, R.P., 2020. A Gradient Flow Framework For Analyzing Network Pruning. arXiv e-prints, pp.arXiv-2009.

---

> > > ### Author Response · Authors · 2020-11-25
> > > **Thanks for the thorough review! - Reply 2 (Part I/I)**
> > >
> > > We thank R1 for this opportunity to address more of your previous concerns.
> > >
> > > ----->“While it is interesting to compare sparse networks to their same capacity dense networks, this comparison is less practical because small dense networks are rarely used in practice.”
> > >
> > > Our goal is to have a controlled experimental framework where we can measure what aspects of the architecture disproportionately impact sparse networks. However, we see this as a framework that generalizes to where this constraint is not in place. For example, we show the phenomena studied in SC-SDC is also present in WideResnet-50 when we ensure they have the similiar network configuration (section 4.2).
> > >
> > > ----->“How are the sparse connections determined? If it is random, then the findings are less applicable because people would use sparse networks found by a good pruning method rather than a random sparse network in practice.”
> > >
> > > We thank R1 for this comment. Our experiments are indeed done on random, sparse networks. We consider this to be a lower bound on expected performance, and thus the most conservative measurement of performance. We have clarified this in our latest version (paragraph 4 in the introduction). In future, we will explore if these findings generalize to pruned networks.
> > >
> > >
> > > ----->“While table 1 shows that EGF is better than simple gradient norms, gradient flow measures in general do not seem that important to the rest of the paper: why do we need gradient flow when we can just look at accuracy? Is EGF used for anything besides as an observation? Also, it is unclear in figure 2 if high EGF can explain good performance or if they are just somewhat correlated.”
> > >
> > > We have extended our experiments and analysis to show the benefits of EGF, namely:
> > > - For adaptive optimization methods that use an exponentially decaying average of past squared gradients, such as Adam and RMSProp, a high EGF correlates to poor performance. We can see this with weight decay (with and without batch normalization) and data augmentation ( without batch normalization) as discussed in section 4.1.
> > > - We have included a section of EGF’s favourable properties on page 4. Notably, it can extend current gradient flow analysis methods.
> > >
> > >
> > > ----->Doesn’t data augmentation hurt dense networks more for adam?
> > > We thank R1 for pointing this out. Although, DA does appear to hurt dense networks more for Adam, this difference was not statistically significant. We have clarified this better in the updated manuscript in section 4.1
> > >
> > > ----->How important is EGF and comparing sparse to same size dense.
> > >
> > > Gradient flow (gradient norms) have been used to study network behaviour and as gradient preserving pruning criteria [1,2,3]. Although, top-line metrics such as accuracy can tell us which training methods achieve better performance, they don’t provide possible information as to why certain methods work/fail. We introduced EGF to provide a measure that could possibly explain why certain interventions hurt or improve sparse network performance. This was the case in the analysis of weight decay and data augmentation in section 4.1.
> > >
> > > Furthermore, we also introduced SC-SDC to allow us to disentangle which of this behaviour is a direct result of a network being sparse and to identify what optimization and regularization components need to be adapted for sparse networks.
> > >
> > > References:
> > > [1] Evci, U., Ioannou, Y.A., Keskin, C. and Dauphin, Y., 2020. Gradient Flow in Sparse Neural Networks and How Lottery Tickets Win. arXiv preprint arXiv:2010.03533.
> > >
> > > [2] Wang, C., Zhang, G. and Grosse, R., 2020. Picking winning tickets before training by preserving gradient flow. arXiv preprint arXiv:2002.07376.
> > >
> > > [3] Singh Lubana, E. and Dick, R.P., 2020. A Gradient Flow Framework For Analyzing Network Pruning. arXiv e-prints, pp.arXiv-2009.

---

### Decision · Program_Chairs · 2021-01-07
**Final Decision**

**Decision:**

Reject

**Comment:**

This paper proposed a new measure of effective gradient flow (EGF), and also compared sparse vs. dense networks on CIFAR-10 and CIFAR-100. The notion of EGF would be interesting, but the paper did not present enough evidence to support this notion.